# Information-theoretically Safe Bias Classifier Against Adversarial Attacks

## Abstract

Deep learning has become the cornerstone of recent advances in artificial intelligence. However, the presence of adversarial samples makes deep learning vulnerable in safety-critical applications. Moreover, adversarial examples have been shown to be unavoidable in some degree. So how to keep the network secure under adversarial attacks? To address this issue, we propose the bias classifier. This approach employs the bias component of a neural network, using ReLU as its activation function, as a classifier. The bias classifier has been shown to universally approximate any classification problem with a high degree of probability. Moreover, it can be made information-theoretically safe against the original model gradient-based attack in the sense that any such attack produces a completely random attacking direction for any given input. Thus, the bias classifier provably achieves the maximum possible robust accuracy under specified attacks. Experiments are used to validate our theoretical results and to show that the bias classifier is accurate and robust for simple models.

## 1 Introduction

Deep learning has become the most powerful machine learning method, which has been successfully applied in computer vision, natural language processing, and many other fields. However, the existence of adversarial samples (Szegedy, 2013; Biggio et al., 2013) makes deep learning vulnerable in safety-critical applications such as autonomous driving (Cao et al., 2019), face recognition (Dong et al., 2019), and medical image processing (Ma et al., 2021). Although many effective methods to defend adversaries have been proposed (Xu et al., 2020), adversarial examples have been shown to be inevitable in certain sense (Azulay & Weiss, 2019; Shafahi et al., 2019a; Bastounis et al., 2021; Yu et al., 2023; Gao et al., 2022).

In (Shafahi et al., 2019a), it is shown that for certain data distributions and network $\mathcal{F}$, if $\mathcal{F}$ gives the correct result for a benign sample $x$, then $x$ must have an adversarial example. In (Bastounis et al., 2021), it is proven that for any network $\mathcal{F}$ with a fixed structure, there exists a dataset $\mathcal{D}$ such that if $\mathcal{F}$ is accurate on $\mathcal{D}$ then $\mathcal{F}$ has adversarial examples. In (Yu et al., 2023), it is proven that if the network is large enough, then small perturbation of the network parameters will lead to adversarial examples. On the other hand, for a given dataset $\mathcal{D}$, it is shown that there exists a network which is robust for $\mathcal{D}$ with a given budget (Li et al., 2022a; Yu et al., 2024), but the network is too large to be practical. Furthermore, in order to be robustly generalizable, the network must have an exponential number of parameters (Li et al., 2022a).

There are many methods to improve the robustness of networks (Xu et al., 2020). Adversarial training (Madry, 2017) is one of the most used defenses and has been shown to be optimal in certain sense (Gao et al., 2022). However, the robust accuracy for networks developed through adversarial training is not very high for moderately complex dataset such as CIFAR-10. Besides, the application of diffusion models for input purification achieves good results to some extent (Nie et al., 2022). But all these defenses cannot give the provable highest robust accuracy.

Since theoretical and experimental results show that adversarial examples cannot be totally eliminated for the commonly used network structures and the practical robust accuracies do not achieve provable highest values, we may ask the following basic question about the neural network security.

**Question. For neural networks with a given structure, for instance VGG-19 or ResNet-18, is it possible to provably achieve the highest possible robust accuracy under certain attacks?**

In this paper, we present a partial answer to the above question by introducing the bias classifier, that is, using the bias part of a deep neural network (DNN) as the classifier. We show that the bias classifier can be made information-theoretically safe for certain attacks in the sense that the success rate of adversarial attack is close to that of a random direction attack. Since random direction attack can be considered the weakest attack, robust accuracy under this attack is the *highest possible*, and in this sense, the information-theoretically safe bias classifier gives the highest robust accuracy.

Let $\mathbb{I} = [0, 1]$ and $\mathcal{F} : \mathbb{I}^n \to \mathbb{R}^m$ be a classification DNN and $m$ the number of labels, using ReLU as the activation function. For any $x \in \mathbb{I}^n$, there exist $W_x \in \mathbb{R}^{m \times n}$ and $B_x \in \mathbb{R}^m$ such that

$$\mathcal{F}(x) = W_x x + B_x.$$

By the *bias classifier*, we mean to use $\mathcal{F}$'s bias part $\mathcal{B}_{\mathcal{F}}(x) = B_x : \mathbb{I}^n \to \mathbb{R}^m$ as a classifier. The contributions of this paper are summarized below.

First, the universal classification power of the bias classifier is proved. Precisely, it is shown that for any classification problem $G : \mathbb{I}^n \to [m]$, there exists a DNN whose bias part gives the correct label $G(x)$ for $x \in \mathbb{I}^n$ with arbitrarily high probability.

Second, the bias classifier can be provably safe against certain gradient-based attacks of the form: $\mathcal{A}(x, y) = x + \rho D_{\mathcal{A}}(x, y) : \mathbb{I}^n \to \mathbb{I}^n$, where $\rho \in \mathbb{R}_+$ is the attack coefficient and $D_{\mathcal{A}}(x, y) \in \{-1, 1\}^n$ is the attack direction. A neural network $\mathcal{F} : \mathbb{I}^n \to \mathbb{R}^m$ is called *information-theoretically safe* against an attack $\mathcal{A}$, if a random attack direction $D_{\mathcal{A}}(x, y)$ is generated for any input sample $x$. The robust accuracy under such an attack $\mathcal{A}$ is equal to the robust accuracy under an attack using the random attack direction. It is clear that such an attack gives the highest possible robust accuracy, which can be achieved practically. This seems to be the first time that the concept of information-theoretically safe classifier is proposed for adversarial attacks. The notion of information-theoretically safe is borrowed from cryptography (Goldreich, 2001), which means that the ciphertext yields no information regarding the plaintext for cyphers that are perfectly random.

Third, an effective training method for the bias classifier is proposed. It is observed that the adversarial training introduced in (Madry, 2017) significantly increases the classification power of the bias part. Using the adversarial training to the loss function $L_{\text{CE}}(\mathcal{B}_{\mathcal{F}}(x), y) + \gamma L_{\text{CE}}(\mathcal{F}(x), y)$ further increases the classification power of the bias part, and hence is used to train the bias classifier.

Finally, experimental results are used to validate the theoretical results and to show that the bias classifier is quite accurate and robust on some simple tasks. We first use experiments to show that the bias classifier is indeed informtion-theoretically safe by achieving the highest possible robust accuracy. We then use simple networks such as Lenet-5 for MNIST and VGG-19 for CIFAR-10 to illustrate the practical performance of the bias classifier. Following (Carlini et al., 2019), we show that the bias classifier is quite robust against major attacks, such as gradient-based white box attack, black box attack, AutoAttack, and BDPA attack (Athalye et al., 2018).

## 2 RELATED WORK

**Adversarial Defenses.** There exist two main approaches to obtain more robust DNNs: using a better training method or a better structured DNN. Of course, the two approaches can be combined.

Many new defensive structures for DNNs were proposed (Xu et al., 2020). Closely related to this paper, gradient masking or obfuscation defenses try to make the gradient information of the DNN unuseful, so that attacks based on gradient cannot work directly (Athalye et al., 2018). Gradient masking methods include the shattered gradients (Buckman et al., 2018; Guo et al., 2017), the stochastic gradients (Dhillon et al., 2018; Xie et al., 2017), the exploding gradients (Song et al., 2017; Samangouei, 2018), and the quantization methods (Bernhard et al., 2019; Khalid et al., 2019; Song et al., 2021). In (Athalye et al., 2018), it was shown that most of these defences can be circumvented by properly learning the gradients. Other models were also introduced to defend adversarial samples, including the ensemble adversarial training (Tramèr et al., 2017), denoising layer (Xie et al., 2019), and difference-privacy noise layer (Lecuyer et al., 2019),

The bias classifier introduced in this paper is similar to gradient obfuscation defense, but is essentially different and has the following novel properties. First, the bias classifier has zero gradient, unlike other methods, which try to make the gradients approximately zero. Second, we show that BPDA attack cannot break the bias classifier in our experimental settings, while the BPDA attack can break most gradient obfuscated defences (Athalye et al., 2018). Finally, the bias classifier can be made *provably robust* against the original model gradient-based attack. Previous random gradient obfuscation models do not claim provably robustness.

Many effective methods have been proposed to train more robust DNNs to defend adversarial attacks (Xu et al., 2020; Shafahi et al., 2019b; Zhang et al., 2019; Wu et al., 2020; Nie et al., 2022). Adversarial training (Madry, 2017) is one of the best practical defenses and can achieve optimal adversarial accuracy in certain sense (Gao et al., 2022). In this paper, adversarial training (Madry, 2017) is used for a new loss function to train the bias classifier.

**Provable Robustness.** Provable or certified robustness for DNNs can be achieved by verification or by deriving explicit robust radius bounds or boundaries. Certified verification aims to decide whether a trained DNN is robust by computing the robust boundaries (Li et al., 2023; Wang et al., 2021; Li et al., 2022b). Explicit lower bounds for robust radius are given for DNNs trained with various approaches such as randomized smoothing (Cohen et al., 2019) or $L_2$ regulations (Yu & Gao, 2023). In (Hein & Andriushchenko, 2017; Raghunathan et al., 2018), some security boundaries were given. Compared to the above approaches, our approach is more practical in that our models are trained like the usual adversarial training, while certified verification usually requires heavy computation, such as SMT solving (Katz et al., 2017) and only works for small DNNs, and the explicit robust radius is usually very small (Yu & Gao, 2023) to be useful for commonly used DNN models.

## 3 BIAS CLASSIFIER

In this section, we define the bias classifier and prove the universal power of the bias classifier. We also provide a training algorithm for the bias classifier.

### 3.1 BIAS CLASSIFIER

Let $\mathbb{I} = [0,1]$ and $[n] = \{1,\ldots,n\}$. For $n, m, L \in \mathbb{N}_+$, let $\mathcal{F} : \mathbb{I}^n \to \mathbb{R}^m$ be a classification DNN with $L$ hidden layers. Each hidden layer of $\mathcal{F}$ uses ReLU as activity functions, and the output layer is an affine transformation. We write $\mathcal{F} : \mathbb{I}^n \to \mathbb{R}^m$ as

$$\mathcal{F}(x_0) = W_{L+1} x_L + b_{L+1} \text{ and } x_l = \text{ReLU}(W_l x_{l-1} + b_l) \in \mathbb{R}^{n_l}, l \in [L], \tag{1}$$

where $x_0 \in \mathbb{I}^n, n_0 = n, n_{L+1} = m, W_l \in \mathbb{R}^{n_l \times n_{l-1}}, b_l \in \mathbb{R}^{n_l}$. Denote $\Theta_{\mathcal{F}} = \{W_l, b_l\}_{l=1}^{L+1}$ to be the parameter set of $\mathcal{F}$. For a label $l \in [m]$ and $x \in \mathbb{I}^n$, denote by $\mathcal{F}^{(l)}(x)$ the $l$-th coordinate of $\mathcal{F}(x)$. The classification result of $\mathcal{F}$ is $\widehat{\mathcal{F}}(x) = \text{argmax}_{l \in [m]} \mathcal{F}^{(l)}(x)$.

Let $\mathcal{F}$ be a network defined as before, for any $x \in \mathbb{I}^n$, we have

$$\mathcal{F}(x) = W_{\mathcal{F}}(x) + \mathcal{B}_{\mathcal{F}}(x) = W_x x + B_x, \tag{2}$$

where $W_x = \mathcal{J}_{\mathcal{F}}(x) = \frac{\nabla \mathcal{F}(t)}{\nabla t}|_x$ is the Jacobian of $\mathcal{F}$ at $x$ and $\mathcal{B}_{\mathcal{F}}(x) = B_x$ is the bias of $\mathcal{F}$ at $x$. We will let the bias part $\mathcal{B}_{\mathcal{F}} : \mathbb{I}^n \to \mathbb{R}^m$ be used as a classifier and is called the *bias classifier*, which can be computed from $\mathcal{F}$ as follows:

$$\mathcal{B}_{\mathcal{F}}(x) = \mathcal{F}(x) - W_{\mathcal{F}}(x) = \mathcal{F}(x) - \frac{\nabla \mathcal{F}(t)}{\nabla t}|_x \cdot x = \mathcal{F}(x) - \mathcal{J}_{\mathcal{F}}(x) \cdot x. \tag{3}$$

**Remark 1.** *Due to the property of the ReLU function, $\mathcal{F}$ is a piecewise linear function. A linear region of $\mathcal{F}$ is a closed maximal connected subset of the input space $\mathbb{I}^n$, on which $\mathcal{F}$ is linear. On each linear region $A$ of $\mathcal{F}$, there exist $W_A \in \mathbb{R}^{m \times n}$ and $B_A \in \mathbb{R}^m$, such that $\mathcal{F}(x) = W_A x + B_A$ for $x \in A$. Obviously, $W_A = W_x$ and $B_A = B_x$ for $x \in A$. It is clear that there exists a finite number of linear regions and $\mathbb{I}^n$ is the union of these linear regions (Goodfellow et al., 2014). As a consequence, the bias classifier is a piecewise constant function with a finite number of values.*

## 3.2 BIAS CLASSIFIER HAS UNIVERSAL POWER FOR CLASSIFICATION

We will prove that for any classification function $G : \mathbb{I}^n \to [m]$, there exists a bias classifier $\mathcal{B}_{\mathcal{F}}(x) : \mathbb{I}^n \to \mathbb{R}^m$ such that $\widehat{\mathcal{B}}_{\mathcal{F}}(x)$ can be arbitrarily close to $G$, where $\widehat{\mathcal{B}}_{\mathcal{F}}(x)$ is the classification result of $\mathcal{B}_{\mathcal{F}}(x)$. Specifically, we define classification functions as follows (Cybenko, 1989).

**Definition 2.** $G : \mathbb{I}^n \to [m]$ *is called a classification function, if* $\mathbb{I}^n = \cup_{i=1}^m P_i$ *is a partition of* $\mathbb{I}^n$ *into* $m$ *disjoint and measurable subsets and* $G(x) = i$ *if and only if* $x \in P_i$.

**Remark 3.** *Real-world classification problems such as MNIST or CIFAR-10 can be treated as classification functions as follows. Let* $\mathbb{O} \subset \mathbb{I}^n$ *be the objects to be classified. For* $x \in \mathbb{O}$ *and* $r \in \mathbb{R}_+$, *when* $r$ *is small enough, it is reasonable to treat all objects in*

$$\mathbb{B}(x, r) = \{x + \eta \,|\, \eta \in \mathbb{R}^n, ||\eta||_\infty < r\}$$

*as images that have the same label as* $x$. *Furthermore, for all elements in* $\mathbb{I}^n \setminus \cup_{x \in \mathbb{O}} \mathbb{B}(x, r)$, *we may assign a new label. In this way, all elements of* $\mathbb{I}^n$ *have a label.*

The following existence theorem shows that the bias classifier has the power to interpolate any classification functions with arbitray high probability.

**Theorem 4.** *Let* $G : \mathbb{I}^n \to [m]$ *be a classification function. Then for any* $\epsilon \in (0, 1/2)$, *there exists a neural network* $\mathcal{F} : \mathbb{I}^n \to \mathbb{R}^m$ *and an open set* $D \subset \mathbb{I}^n$ *with volume* $V(D) < \epsilon$, *such that* $\widehat{\mathcal{B}}_{\mathcal{F}}(x, y) = G(x)$ *for* $x \in \mathbb{I} \setminus D$.

**Proof Idea**. By the universal approximation theorem given in (Cybenko, 1989), we can construct a neural network $\mathcal{G} : \mathbb{I}^n \to \mathbb{R}^m$ with a step activation function that can approximate $G(x)$. Furthermore, we show that $\mathcal{G}$ can be modified such that all entries in its bias are nonzero. Furthermore, a neural network $\mathcal{F} : \mathbb{I}^n \to \mathbb{R}^m$ with ReLU as the activation function can be obtained from $\mathcal{G}$, which satisfies the condition of Theorem 4. Details of the proof are given in Appendix A.

## 3.3 TRAINING THE BIAS CLASSIFIER

Given a training set $\mathcal{S} \subset \mathbb{I}^n \times [m]$, normal training is to solve the following optimization problem:

$$\min_{\mathcal{F}} \sum_{(x,y) \in \mathcal{S}} L_{\text{CE}}(\mathcal{F}(x), y). \tag{4}$$

Adversarial training (Madry, 2017) is to solve the following optimization problem:

$$\min_{\mathcal{F}} \sum_{(x,y) \in \mathcal{S}} \max_{||\zeta|| < \varepsilon} L_{\text{CE}}(\mathcal{F}(x + \zeta), y) \tag{5}$$

where $\varepsilon \in \mathbb{R}_+$ is a given small number. As shown in table 1, adversarial training helps to improve the power of the bias classifier. In order to further increase the power of the bias classifier $\mathcal{B}_{\mathcal{F}}$ and to keep the training procedure efficient to update the parameters, we use the following loss function to train the bias classifier

$$\min_{\mathcal{F}} \sum_{(x,y) \in \mathcal{S}} [L_{\text{CE}}(\mathcal{B}_{\mathcal{F}}(x + \zeta_{x,y,\mathcal{F}}), y) + \gamma L_{\text{CE}}(\mathcal{F}(x + \zeta_{x,y,\mathcal{F}}), y)]$$

$$\zeta_{x,y,\mathcal{F}} = \text{argmax}_{||\zeta|| < \varepsilon} L_{\text{CE}}(\mathcal{F}(x + \zeta), y) \tag{6}$$

where $\gamma \in \mathbb{R}_+$ is a hyperparameter. This loss function tries to add the performance of bias classifier in the adversarial data into the loss function. Because bias classifier does not have gradients, it is difficult to quickly find adversarial samples by using gradients, so we use the adversarial samples from the original network as a substitution. Please note that the parameters have derivatives for each part of equation 6 , so training can proceed.

By Table 1, easy to see that adversarial training for the bias classifier in equation 6 further increases the power of the bias part.

Table 1: Accuracies of network Lenet-5 for MNIST on the test set.

| Training Methods | $\mathcal{B}_{\mathcal{F}}$ | $\mathcal{F}$ |
|---|---|---|
| Normal training equation 4 | 15.62% | 99.09% |
| Adversarial training equation 5 | 98.77% | 99.19% |
| Adversarial training for bias classifier equation 6 | 99.09 % | 99.43% |

## 4 INFORMATION-THEORETICALLY SAFE BIAS CLASSIFIER

In this section, we will first define a common attack method of bias classifier and **Information-theoretically safe**, and then build a information-theoretically safe bias classifier based on such attack methods and prove it.

### 4.1 ORIGINAL MODEL GRADIENT-BASED ATTACK

The most popular methods to generate adversarial examples for network $\mathcal{F}$, such as FGSM (Goodfellow et al., 2014) or PGD (Madry, 2017), use gradient $\frac{\nabla \mathcal{F}(x)}{\nabla x}$ to make the loss function larger. For example, adversarial examples are generated by FGSM as follows

$$x \to x + \rho \operatorname{sign}(\frac{\nabla L_{\mathrm{CE}}(\mathcal{F}(x), y)}{\nabla x}) \tag{7}$$

for a small parameter $\rho \in \mathbb{R}_+$. Motivated by this fact, we introduce the concept of gradient-based attack. We mainly consider one-step attack , the multistep attacks are discussed in Appendix D.

**Definition 5.** *Let $\mathcal{F} : \mathbb{I}^n \to \mathbb{R}^m$ be a network, $\rho \in \mathbb{R}_+$, and $D_{\mathcal{F}}(x, y) : \mathbb{I}^n \times \mathbb{R} \to \mathbb{R}^n$. Then*

$$\mathcal{A}_{\mathcal{F}, D_{\mathcal{F}}}(x, y) = x + \rho\, D_{\mathcal{F}}(x, y) : \mathbb{R}^{n+1} \to \mathbb{R}^n \tag{8}$$

*is called a* gradient-based attack *for $\mathcal{F}$ if the attack direction $D_{\mathcal{F}}(x, y)$ depends only on the sample $(x, y)$, the network output $\mathcal{F}(x)$, and the gradient $\frac{\nabla \mathcal{F}(x)}{\nabla x}$.*

For example, the FGSM attack is a gradient-based attack based on attack direction $D_{\mathcal{F}}(x, y) = \operatorname{sign}(\frac{\nabla L_{CE}(\mathcal{F}(x), y)}{\nabla x})$, because $\frac{\nabla L_{CE}(\mathcal{F}(x), y)}{\nabla x}$ can be calculated by $(x, y)$, $\mathcal{F}(x)$ and $\frac{\nabla \mathcal{F}(x)}{\nabla x}$.

Since the gradient of $\mathcal{B}_{\mathcal{F}}$ is zero, we cannot directly use a gradient-based attack on $\mathcal{B}_{\mathcal{F}}$, but an obvious alternative attack to $\mathcal{B}_{\mathcal{F}}$ is to use the gradients of $\mathcal{F}$, which leads to the following definition.

**Definition 6.** *$\mathcal{F}$ and $D_{\mathcal{F}}(x, y)$ are following Definition 5. Then*

$$\mathcal{A}_{\mathcal{B}_{\mathcal{F}}, D_{\mathcal{F}}}(x, y) = x + \rho\, D_{\mathcal{F}}(x, y) : \mathbb{R}^{n+1} \to \mathbb{R}^n \tag{9}$$

*is called an* original model gradient-based attack *for $\mathcal{B}_{\mathcal{F}}$ based on* attack direction $D_{\mathcal{F}}$.

### 4.2 INFORMATION-THEORETICALLY SAFETY

In this section, we define the concept of information-theoretically safety. Let $\mathcal{F}_R$ be a neural network that involves a random variable $R$ satisfying the distribution $\mathcal{R}$. Safety of $\mathcal{F}_R$ against an attack $\mathcal{A}$ can be measured by the following *adversarial creation rate*:

$$\mathcal{C}(\mathcal{F}_R, \mathcal{A}, \mathcal{R}) = \mathbb{E}_{R \sim \mathcal{R}}[\mathbb{E}_{(x, y) \sim \mathcal{D}}[\mathbf{I}(\widehat{\mathcal{F}}_R(\mathcal{A}(x, y)) \neq \widehat{\mathcal{F}}_R(x))]] \tag{10}$$

where $\mathcal{D}$ is the data distribution. Attack $\mathcal{A}$ is called a *random direction attack* if $\mathcal{A}(x, y)$ is a random direction in $\{-1, 1\}^n$ for any $(x, y) \sim \mathcal{D}$.

**Definition 7.** *The above mentioned neural network $\mathcal{F}_R$ is called* information-theoretically safe *against an attack $\mathcal{A}$, if*

$$\mathcal{C}(\mathcal{F}_R, \mathcal{A}, \mathcal{R}) \leq \inf_{R \sim \mathcal{R}} \mathcal{C}(\mathcal{F}_R), \tag{11}$$

*where $\mathcal{C}(\mathcal{F}_R) = \frac{1}{2^n} \sum_{V \in \{-1, 1\}^n} \mathbb{E}_{(x, y) \sim \mathcal{D}}[\mathbf{I}(\widehat{\mathcal{F}}_R(x + \rho V) \neq \widehat{\mathcal{F}}_R(x))]$ is the creation rate of adversarial examples for random direction attack against network $\mathcal{F}_R$.*

We now show how to build an information-theoretically safe bias classifier under original model gradient-based attack mentioned in Section 4.1. First train a network $\mathcal{F} : \mathbb{I}^n \to \mathbb{R}^m$ with the method in Section 3.3. Let $W_R$ be sampled from a distribution $\mathcal{M}$ of random matrices in $\mathbb{R}^{m,n}$ and let

$$\begin{aligned} \overline{\mathcal{F}}(x) &= \mathcal{F}(x) + W_R x = (W_x + W_R)x + B_x \\ \mathcal{B}_{\overline{\mathcal{F}}}(x) &= \overline{\mathcal{F}}(x) - \frac{\nabla \overline{\mathcal{F}}(x)}{\nabla x} \cdot x = \overline{\mathcal{F}}(x) - \mathcal{J}_{\overline{\mathcal{F}}} \cdot x. \end{aligned} \tag{12}$$

It is easy to see that $\mathcal{B}_{\overline{\mathcal{F}}} = \mathcal{B}_{\mathcal{F}}$, that is, the bias classifiers for $\mathcal{F}$ and $\overline{\mathcal{F}}$ are the same. Therefore, we have

$$\inf_{W_R \sim \mathcal{M}} \mathcal{C}(B_{\overline{\mathcal{F}}}) = \mathcal{C}(B_{\mathcal{F}}). \tag{13}$$

**Remark 8.** $\mathcal{C}(B_{\mathcal{F}})$ *is the creation rate of adversarial examples for random direction attack against* $B_{\mathcal{F}}$, *which is always small for common networks $\mathcal{F}$ as shown in Table 3. Thus, if $B_{\overline{\mathcal{F}}}$ is information-theoretically safe against attack $\mathcal{A}$, then the adversarial accuracy under attack $\mathcal{A}$ is equal to that of random direction attack, which is the highest possible value for the adversarial accuracy. If $B_{\overline{\mathcal{F}}}$ is not information-theoretically safe against $\mathcal{A}$, then we can use the value $\mathcal{C}(B_{\overline{\mathcal{F}}}, \mathcal{A}, \mathcal{M})/\mathcal{C}(B_{\mathcal{F}})$ to measure the safety of $B_{\overline{\mathcal{F}}}$ relative to the information-theoretically safety.*

**Remark 9.** *When $m$ or $n$ is large, $W_R$ contains many parameters. This is not a problem. First, $W_R$ is generated randomly and does not need to be trained. Second, in practice, we can introduce randomness using fewer parameters. For example, we can first generate a CNN $\mathcal{F}_R \in \mathbb{I}^n \to \mathbb{R}^m$ without bias part and with random weights and then use $\mathcal{F}_R(x)$ instead of $W_R x$ in equation 12.*

### 4.3 INFORMATION-THEORETICALLY SAFETY AGAINST CARLINI-WAGNER ATTACK

We first consider the *Carlini-Wagner attack* (Carlini & Wagner, 2017) $\mathcal{A}_{\mathcal{B}_{\mathcal{F}}, \mathcal{D}_{\mathcal{F}}^1}$, which is a gradient-based attack using the following attack direction:

$$\mathcal{D}_{\mathcal{F}}^1(x, y) = \text{sign}\left(\frac{\nabla \mathcal{F}_{n_x}(x)}{\nabla x} - \frac{\nabla \mathcal{F}_y(x)}{\nabla x}\right) \tag{14}$$

where $n_x = \arg\max_{i \neq y}\{\mathcal{F}^{(i)}(x)\}$.

Denote $||W||_{\infty} = \max_{i,j}|w_{i,j}|$ for $W \in \mathbb{R}^{m \times n}$ and denote $||x||_{-\infty} = \min_{i \in [n]}\{|x_i|\}$ for $x \in \mathbb{R}^n$. We consider two types of random matrix for $W_R$ in equation 12 to be defined below.

**Definition 10.** *Let $\mathcal{I}(a, b)$ be the uniform distribution in $[a, b] \subset \mathbb{R}$. For $\lambda \in \mathbb{R}_+$, denote $\mathcal{U}_{m,n}(\lambda)$ to be the random matrices whose entries are in $\mathcal{I}(-\lambda, \lambda)$ and denote $\mathcal{M}_{m,n}(\lambda)$ to be the random matrices such that the entries of its $i$-row are in $(\mathcal{I}(-2i\lambda, -(2i-1)\lambda) \cup \mathcal{I}((2i-1)\lambda, 2i\lambda))^n$.*

For the distribution $\mathcal{M}_{m,n}(\lambda)$, the following theorem shows that $\mathcal{B}_{\overline{\mathcal{F}}}$ is information-theoretically safe for attack $\mathcal{A}_{\mathcal{B}_{\overline{\mathcal{F}}}, \mathcal{D}_{\overline{\mathcal{F}}}^1}$ if $\lambda$ is large.

**Theorem 11.** *For $\lambda \in \mathbb{R}_+$, if $||J_{\mathcal{F}}||_{\infty} < \lambda/2$ and $W_R \sim \mathcal{M}_{m,n}(\lambda)$, then $\mathcal{B}_{\overline{\mathcal{F}}}$ in equation 12 is information-theoretically safe against attack $\mathcal{A}_{\mathcal{B}_{\overline{\mathcal{F}}}, \mathcal{D}_{\overline{\mathcal{F}}}^1}$.*

*Proof.* From equation 2 and equation 12, $J_{\overline{\mathcal{F}}} = \frac{\nabla \overline{\mathcal{F}}(x)}{\nabla x} = W_x + W_R$. Let $W_{R,i}$ and $W_{x,i}$ be the $i$-rows of $W_R$ and $W_x$, respectively. Since $W_R \sim \mathcal{M}_{m,n}(\lambda)$, we have $||W_{R,i} - W_{R,j}||_{-\infty} > \lambda$ for $i \neq j$. Since $||\mathcal{F}||_{\infty} = ||\frac{\nabla \mathcal{F}(x)}{\nabla x}||_{\infty} = |W_x|_{\infty} < \lambda/2$, we have $||W_{x,i} - W_{x,j}||_{\infty} < \lambda$ for $i \neq j$. From $||W_{R,i} - W_{R,j}||_{-\infty} > \lambda$ and $||W_{x,i} - W_{x,j}||_{\infty} < \lambda$ for $i \neq j$, we have

$$\begin{aligned}
\mathcal{A}_{\mathcal{B}_{\overline{\mathcal{F}}}, \mathcal{D}_{\overline{\mathcal{F}}}^1}(x) &= x + \rho\,\text{sign}\left(\frac{\nabla \overline{\mathcal{F}}_{n_x}(x)}{\nabla x} - \frac{\nabla \overline{\mathcal{F}}_y(x)}{\nabla x}\right) \\
&= x + \rho\,\text{sign}(W_{x,n_x} - W_{x,y} + W_{R,n_x} - W_{R,y}) \\
&= x + \rho\,\text{sign}(W_{R,n_x} - W_{R,y}).
\end{aligned} \tag{15}$$

Since $W_R \sim \mathcal{M}_{m,n}(\lambda)$, $\widehat{W} = W_{R,n_x} - W_{R,y}$ is a random vector whose entries are in $[-b_2, -b_1] \cup [b_1, b_2]$ for some $b_1, b_2 \in \mathbb{R}$, and thus we have that:

$$\begin{aligned}
\mathcal{C}(\mathcal{B}_{\overline{\mathcal{F}}}, \mathcal{A}_{\mathcal{B}_{\overline{\mathcal{F}}}, \mathcal{D}_{\overline{\mathcal{F}}}^1}, \mathcal{M}_{m,n}(\lambda)) &= \mathbb{E}_{W_R \sim \mathcal{M}_{m,n}(\lambda)}[\mathbb{E}_{x \sim \mathcal{D}}[\mathbf{I}(\widehat{\mathcal{B}}_{\overline{\mathcal{F}}}(\mathcal{A}_{\mathcal{B}_{\overline{\mathcal{F}}}, \mathcal{D}_{\overline{\mathcal{F}}}^1}(x)) \neq \widehat{\mathcal{B}}_{\overline{\mathcal{F}}}(x))]] \\
&= \mathbb{E}_{W_R \sim \mathcal{M}_{m,n}(\lambda)}[\mathbb{E}_{x \sim \mathcal{D}}[\mathbf{I}(\widehat{\mathcal{B}}_{\overline{\mathcal{F}}}(x + \rho\,\text{sign}(W_R)) \neq \widehat{\mathcal{B}}_{\overline{\mathcal{F}}}(x))]] \\
&= \frac{1}{2^n}\sum_{V \in \{-1,1\}^n} \mathbb{E}_{x \sim \mathcal{D}}[\mathbf{I}(\widehat{\mathcal{B}}_{\mathcal{F}}(x + \rho\,V) \neq \widehat{\mathcal{B}}_{\mathcal{F}}(x))] \\
&= \mathcal{C}(\mathcal{B}_{\mathcal{F}}) = \inf_{W_R \sim \mathcal{M}_{m,n}(\lambda)} \mathcal{C}(B_{\overline{\mathcal{F}}}),
\end{aligned}$$

where equation 13 is used in the last equation. The theorem is proved. $\square$

**Remark 12.** *The original model gradient-based attack requires that only the values of the network and its gradient are known, and its structure is kept as a secret. This is a very strong condition. On the other hand, the reward is also very high: the maximum possible adversarial accuracy can be provably achieved for commonly used networks and attacks. Networks possessing such strong security attributes could be employed in operations demanding assured safety, under strict condition.*

For the simpler distribution $\mathcal{U}_{m,n}(\lambda)$, the following theorem shows that $\mathcal{B}_{\overline{\mathcal{F}}}$ is approximately safe for attack $\mathcal{A}_{\mathcal{B}_{\overline{\mathcal{F}}}, \mathcal{D}_{\overline{\mathcal{F}}}^1}$ under certain conditions. The proof of the theorem is given in Appendix B.

**Theorem 13.** *If $||\mathcal{J}_{\mathcal{F}}||_\infty < \mu/2$ and $W_R \sim \mathcal{U}_{m,n}(\lambda)$, then $\mathcal{C}(\mathcal{B}_{\overline{\mathcal{F}}}, \mathcal{A}_{\mathcal{B}_{\overline{\mathcal{F}}}, \mathcal{D}^1_{\overline{\mathcal{F}}}}, \mathcal{U}_{m,n}(\lambda)) \le \mathcal{C}(\mathcal{B}_{\mathcal{F}}) + \mu n/\lambda$. Furthermore, if $\lambda > \mu n/(\epsilon \mathcal{C}(\mathcal{B}_{\mathcal{F}}))$, then $\mathcal{C}(\mathcal{B}_{\overline{\mathcal{F}}}, \mathcal{A}_{\mathcal{B}_{\overline{\mathcal{F}}}, \mathcal{D}^1_{\overline{\mathcal{F}}}}, \mathcal{U}_{m,n}(\lambda)) \le (1 + \epsilon)\mathcal{C}(\mathcal{B}_{\mathcal{F}})$.*

By Remark 8, Theorem 13 implies that $\mathcal{B}_{\overline{\mathcal{F}}}$ is close to information-theoretically safe under attack $\mathcal{A}_{\mathcal{B}_{\overline{\mathcal{F}}}, \mathcal{D}^1_{\overline{\mathcal{F}}}}$ if we use a sufficiently large $\lambda$.

### 4.4 Information-theoretically safety against FGSM attack

In this section, we consider the FGSM attack $\mathcal{A}_{\mathcal{B}_{\overline{\mathcal{F}}}, \mathcal{D}^2_{\overline{\mathcal{F}}}}(x)$ (Goodfellow et al., 2014), which is a gradient-based attack using the following attack direction:

$$\mathcal{D}^2_{\mathcal{F}}(x, y) = \text{sign}(\frac{\nabla L_{CE}(\mathcal{F}(x), y)}{\nabla x}). \tag{16}$$

The following theorem shows that $\mathcal{B}_{\overline{\mathcal{F}}}$ in equation 12 is safe against the FGSM attack for binary classification problems. All proofs are given in Appendix C.

**Theorem 14.** *For $\lambda \in \mathbb{R}_+$, if $||J_{\mathcal{F}}||_\infty < \lambda/2$, $W_R \sim \mathcal{M}_{m,n}(\lambda)$, and $m = 2$, then $\mathcal{B}_{\overline{\mathcal{F}}}$ is information-theoretically safe against attack $\mathcal{A}_{\mathcal{B}_{\overline{\mathcal{F}}}, \mathcal{D}^2_{\overline{\mathcal{F}}}}$.*

When $m > 2$, we have the following result.

**Theorem 15.** *Let $||J_{\mathcal{F}}||_\infty < \mu/2$, $||\mathcal{B}_{\mathcal{F}}||_\infty < \beta$, and $\lambda \in \mathbb{R}_+$ satisfy $\lambda > \mu$ and $(\lambda - \mu)e^{-2\beta - n\mu + \sqrt{\lambda}} > (2m\lambda + \mu)m$. Furthermore, assume that the samples are normalized, that is, $||x||_\infty = 1$. If $W_R \sim \mathcal{M}_{m,n}(\lambda)$, then $\mathcal{C}(\mathcal{B}_{\overline{\mathcal{F}}}, \mathcal{A}_{\mathcal{B}_{\overline{\mathcal{F}}}, \mathcal{D}^2_{\overline{\mathcal{F}}}}, \mathcal{M}_{m,n}(\lambda)) \le (m-1)\mathcal{C}(\mathcal{B}_{\mathcal{F}}) + \frac{(m-2)^2}{\sqrt{\lambda}}$.*

**Remark 16.** *We can choose a large $\lambda$ to make the term $\frac{(m-2)^2}{2\sqrt{\lambda}\eta}$ small. From Table 3, $\mathcal{C}(\mathcal{B}_{\overline{\mathcal{F}}})$ is very small for MNIST and CIFAR-10. So, by Theorem 15, $\mathcal{B}_{\overline{\mathcal{F}}}$ is approximately safe if $m$ is small. If $m$ is large, we cannot deduce the safety of $B_{\overline{\mathcal{F}}}$ from Theorem 15. Please note that this does not imply that the bias classifier is not safe, because the upper bound given in Theorem 15 may not be tight, and more research on the upper bound of $\mathcal{C}(\mathcal{B}_{\overline{\mathcal{F}}}, \mathcal{A}_{\mathcal{B}_{\overline{\mathcal{F}}}, \mathcal{D}^2_{\overline{\mathcal{F}}}}, \mathcal{M}_{m,n}(\lambda))$ is needed.*

In the remainder of this section, we consider the safety of $\mathcal{B}_{\overline{\mathcal{F}}}$ when $W_R \sim \mathcal{U}_{m,n}(\lambda)$.

**Theorem 17.** *If $||\mathcal{J}_{\mathcal{F}}||_\infty < \mu/2$, $W_R \sim \mathcal{U}_{m,n}(\lambda)$, and $m = 2$, then $\mathcal{C}(\mathcal{B}_{\overline{\mathcal{F}}}, \mathcal{A}_{\mathcal{B}_{\overline{\mathcal{F}}}, \mathcal{D}^2_{\overline{\mathcal{F}}}}, \mathcal{U}_{m,n}(\lambda)) \le e^{n\mu/\lambda}\mathcal{C}(\mathcal{B}_{\mathcal{F}})$. Furthermore, if $\lambda > n\mu/\ln(1 + \epsilon)$, then $\mathcal{C}(\mathcal{B}_{\overline{\mathcal{F}}}, \mathcal{A}_{\mathcal{B}_{\overline{\mathcal{F}}}, \mathcal{D}^2_{\overline{\mathcal{F}}}}, \mathcal{U}_{m,n}(\lambda)) \le (1 + \epsilon)\mathcal{C}(\mathcal{B}_{\mathcal{F}})$.*

For the general $m$, we have

**Theorem 18.** *Assume $||\mathcal{J}_{\mathcal{F}}||_\infty < \mu/4$, $||\mathcal{B}_{\mathcal{F}}||_\infty < \beta$, and $\lambda \in \mathbb{R}_+$ satisfying $\mu e^{-2\beta - n\mu/2 + \sqrt{\lambda}} > 2(2\lambda + \mu)m$. Furthermore, assume that the samples are normalized, that is, $||x||_\infty = 1$. If $W_R \sim \mathcal{U}_{m,n}(\lambda)$, then $\mathcal{C}(\mathcal{B}_{\overline{\mathcal{F}}}, \mathcal{A}_{\mathcal{B}_{\overline{\mathcal{F}}}, \mathcal{D}^2_{\overline{\mathcal{F}}}}, \mathcal{U}_{m,n}(\lambda)) \le (m-1)\mathcal{C}(\mathcal{B}_{\mathcal{F}}) + \frac{(m-1)n\mu}{\lambda} + \frac{(m-2)^2}{\sqrt{\lambda}}$.*

Theorem 17 shows that, for binary classification, $\mathcal{B}_{\overline{\mathcal{F}}}$ is close to information-theoretically safe against FGSM under distribution $\mathcal{U}_{m,n}(\lambda)$ for sufficiently large $\lambda$. Theorem 18 shows that the result is approximately valid if $m$ is small. Refer to Remark 16 for a more detailed discussion.

## 5 Experiments

In this section, experiments are used to validate the theoretical results and to show that the bias classifier achieves comparable results in terms of accuracy and robustness for simple models.

**Basic settings** We consider the datasets MNIST and CIFAR10, and the following two bias classifiers:

$$\begin{array}{ll} \mathcal{B}_{\mathcal{F}^{(1)}} : \mathcal{F}^{(1)} \text{ is VGG16 for CIFAR10 (Lenet-5 for MNIST), trained with equation 6} \\ \mathcal{B}_{\mathcal{F}^{(2)}} : \mathcal{F}^{(2)} \text{ is ResNet18 for CIFAR10 (VGG9 for MNIST), trained with equation 6} \end{array} \tag{17}$$

We train 200 epoches and learning rate 0.04, halve at 100 and 150 epoches. Use PGD-10 to find adversarial in the training, and attack budget is $L_\infty$ norm 0.1 for MNIST and 8/255 for CIFAR-10,

the same budget is also used in the test after training. The accuracy and adversarial accuracy under original based AutoAttack of these two bias classifiers are shown in table 2, the bias classifier itself does not have good ability to improve robustness, but by using the ideas in equation 12, it can achieve the information-theoretically safety, as shown in the following section.

Table 2: Accuracy(A) and Adversarial Accuracy (AA) on the test sets.

| DNN | MNIST(A) | CIFAR-10(A) | MNIST(AA) | CIFAR-10(AA) |
|---|---|---|---|---|
| $\mathcal{B}_{\mathcal{F}^{(1)}}$ | 99.14% | 81.33% | 98.50% | 31.25% |
| $\mathcal{B}_{\mathcal{F}^{(2)}}$ | 99.06% | 81.64% | 98.03% | 22.54% |

### 5.1 PROVABLE ROBUSTNESS AGAINST ORIGINAL MODEL GRADIENT-BASED ATTACK

In this section, we use experimental results to validate the theoretical results in Section 4.

**Rates of adversaries for random samples** We estimate the adversarial creation rates for random direction attack. In Table 3, we estimate the value of $\mathcal{C}(\mathcal{B}_{\overline{\mathcal{F}}})$ defined in equation 11 for $\rho = 0.1$ (or $\rho = 8/255$ for CIFAR-10), we do it as follows. Randomly take 10000 samples in testset, for each sample, randomly take 1000 random noise in $\{-0.1, 0.1\}^n$ for MNIST ($\{-8/255, 8/255\}^n$ for CIFAR-10) and add them separately to the sample, where $n$ is the dimension of the samples. At last, we estimate the value of $\mathcal{C}(\mathcal{B}_{\overline{\mathcal{F}}})$ on these samples. We can see that $\mathcal{C}(\mathcal{B}_{\mathcal{F}^{(1)}}), \mathcal{C}(\mathcal{B}_{\mathcal{F}^{(2)}})$ are very small, which means that random attacks have little effect in generating adversarial examples. Equivalently, the adversarial accuracy for random direction attack is near 100%.

**Information-theoretically safety of the bias classifier** Following equation 12, let $\mathcal{F}^{(5,i)} = \mathcal{F}^{(i)} + W_5 x$, where $i \in \{1, 2\}$ and $\mathcal{F}^{(i)}$ are given in equation 17, $W_5 \sim \mathcal{U}_{10,784}(100)$ for MNIST, and $W_5 \sim \mathcal{U}_{10,3072}(100)$ for CIFAR-10. Let $\mathcal{F}^{(6,i)} = \mathcal{F}^{(i)} + W_6 x$, $W_6 \sim \mathcal{M}_{10,784}(100)$ for MNIST, and $W_6 \sim \mathcal{M}_{10,3072}(100)$ for CIFAR-10. Following Remark 9, let $\mathcal{F}^{(7,i)} = \mathcal{F}^{(i)} + \mathcal{F}_R$, where $\mathcal{F}_R$ is a CNN with two hidden layers for MNIST and CIFAR-10.

We use original model gradient-based PGD-10 to attack these model and the results are given in Table 5. The results under other attack methods are shown in the Appendix E.

We will show that these bias classifiers almost achieve information-theoretically safety for both MNIST and CIFAR-10. By the definition of information-theoretically safety, we just need to show that the attack success rate of the original model gradient-based attack and random attack are almost the same. By *attack success rate*, we refer to the proportion of samples for which the network gives the correct label but assigns an incorrect label after attack.

According to Table 3, $\mathcal{C}(\mathcal{B}_{\mathcal{F}^{(i)}})$ is also about 1% for MNIST and CIFAR-10, and this is approximately the success rate of the random direction attack. By Table 2, the accuracy of $\mathcal{B}_{\mathcal{F}^{(i)}}$ is about 81% for CIFAR-10 and about 99% for MNIST. By Table 5, the adversarial accuracy of $\mathcal{B}_{\mathcal{F}^{(i)}}$ is about 80% for CIFAR-10 and more than 98% for MNIST. The difference between the corresponding values is about 1%, which means that the attack success rate of the original model gradient-based attack is almost the same as that of the random direction attack. Therefore, information-theoretical safety is achieved.

It is worth mentioning that the bias classifier gives the highest possible adversarial accuracy for MNIST and CIFAR-10. By Table 2, the accuracy of $\mathcal{B}_{\mathcal{F}^{(i)}}$ is about 99% for MNIST and 81% for CIFAR-10. Since the adversarial accuracy is generally less than the accuracy, $\mathcal{B}_{\mathcal{F}^{(i)}}$ achieves the highest possible adversarial accuracy.

Table 3: Value of $\mathcal{C}(\mathcal{B}_{\mathcal{F}^{(1)}}), \mathcal{C}(\mathcal{B}_{\mathcal{F}^{(2)}})$.

| | MNIST | CIFAR-10 |
|---|---|---|
| $\mathcal{C}(\mathcal{B}_{\mathcal{F}^{(1)}})$ | 0.78% | 1.09% |
| $\mathcal{C}(\mathcal{B}_{\mathcal{F}^{(2)}})$ | 0.82% | 1.11% |

Table 4: Adversarial accuracy with the BPDA attack.

| Network | MNIST | CIFAR-10 |
|---|---|---|
| $\mathcal{B}_{\mathcal{F}^{(1)}}$ | 98.18% | 69.51% |
| $\mathcal{B}_{\mathcal{F}^{(2)}}$ | 97.83% | 64.14% |

Table 5: Adversarial accuracy with the original model gradient-based attack.

| Network | MNIST | CIFAR-10 | Network | MNIST | CIFAR-10 |
|---|---|---|---|---|---|
| $\mathcal{B}_{\mathcal{F}^{(5,1)}}$ | 98.67% | 80.15% | $\mathcal{B}_{\mathcal{F}^{(5,2)}}$ | 98.54% | 80.02% |
| $\mathcal{B}_{\mathcal{F}^{(6,1)}}$ | 98.70% | 80.86% | $\mathcal{B}_{\mathcal{F}^{(6,2)}}$ | 98.73% | 80.62% |
| $\mathcal{B}_{\mathcal{F}^{(7,1)}}$ | 98.25% | 79.41% | $\mathcal{B}_{\mathcal{F}^{(7,2)}}$ | 98.44% | 79.48% |

## 5.2 EVALUATION OF THE BIAS CLASSIFIER UNDER GRADIENT-INDEPENDENT ATTACK

The above section shows the information-theoretically safety under the original model gradient-based attack. In this section, we mainly consider the black box attack and the attack against non-gradient networks. We assume that under such attacks of $\mathbb{B}_{\mathcal{F}}$, the attacker can directly obtain the output of $\mathbb{B}_{\mathcal{F}}$. In this case, and consider that for $i = 1, 2$, there are $\mathbb{B}_{\mathcal{F}^{(i)}} = \mathbb{B}_{\mathcal{F}^{(5,i)}} = \mathbb{B}_{\mathcal{F}^{(6,i)}} = \mathbb{B}_{\mathcal{F}^{(7,i)}}$, so we just need to directly consider the attacks on $\mathbb{B}_{\mathcal{F}^i}$.

To compare, we also consider the following networks:

$$
\begin{aligned}
&\mathcal{F}^{(3)} : \text{VGG16 for CIFAR10 (VGG9 for MNIST), trained with equation 5;} \\
&\mathcal{F}^{(4)} : \text{ResNet18 for CIFAR10 (Lenet5 for MNIST), trained with equation 5;} \\
&\mathcal{F}^{(5)} : \text{VGG16 for CIFAR10 (VGG9 for MNIST), trained with TRADES (Zhang et al., 2019);} \\
&\mathcal{F}^{(6)} : \text{ResNet18 for CIFAR10 (Lenet5 for MNIST), trained with TRADES (Zhang et al., 2019).}
\end{aligned}
\tag{18}
$$

**Black-box attacks** We further compare the robustness of $\mathcal{B}_{\mathcal{F}^{(1)}}$, $\mathcal{B}_{\mathcal{F}^{(2)}}$ and $\mathcal{F}^{(i)}, i = 3, 4, 5, 6$ against three black-box attacks: the surrogate model attack, the limited queries and information black-box attack (LQI) (Ilyas et al., 2018), and the zeroth order optimization (ZOO) based black-box attack (Chen et al., 2017). Moreover, to increase the power of the surrogate model attack, for any given bias classifier $\mathbb{B}_{\mathcal{F}}$, three new networks $\overline{\mathcal{F}}_j (j = 1, 2, 3)$ are trained with the data $\{(x, \mathbb{B}_{\mathcal{F}}(x))\}$, and by (Papernot et al., 2017), let $\overline{\mathcal{F}}_j$ have similar structure with $\mathcal{F}$ to increase the transferability: $\overline{\mathcal{F}}_1$ has the same structure with $\mathcal{F}$; $\overline{\mathcal{F}}_2$ is obtained by increasing depth of $\mathcal{F}$; $\overline{\mathcal{F}}_3$ is obtained by increasing the channels of each hidden layer of $\mathcal{F}$. For a sample $x$, create an adversary $x_j$ with $\overline{\mathcal{F}}_j$ for $j = 1, 2, 3$ by AutoAttack, and $x$ is considered to have an adversary if one of $x_j$ is an adversary to $\mathbb{B}_{\mathcal{F}}$.

The results are given in Table 6. For the bias classifier, compared to Table 2, we can see that these black-box attacks are weaker than the white box attack, which is reasonable, and the bias classifier has the defense capability at the same level as other networks.

Table 6: Adversarial accuracy under the black-box attack.

| network | MNIST | | | CIFAR-10 | | |
|---|---|---|---|---|---|---|
| | surrogate model | LQI | ZOO | surrogate model | LQI | ZOO |
| $\mathcal{B}_{\mathcal{F}^{(1)}}$ | 98.81% | 99.01% | 98.15% | 73.51% | 77.20% | 68.15% |
| $\mathcal{B}_{\mathcal{F}^{(2)}}$ | 98.63% | 98.91% | 98.22% | 72.11% | 76.60% | 68.20% |
| $\mathcal{F}^{(3)}$ | 98.52% | 98.94% | 98.49% | 70.85% | 75.21% | 67.17% |
| $\mathcal{F}^{(4)}$ | 98.18% | 98.26% | 98.25% | 69.61% | 74.51% | 68.28% |
| $\mathcal{F}^{(5)}$ | 98.73% | 98.39% | 98.15% | 72.69% | 77.84% | 69.61% |
| $\mathcal{F}^{(6)}$ | 98.46% | 98.68% | 98.32% | 71.48% | 78.35% | 67.68% |

**BPDA attack on the bias classifier** The Backward Pass Differentiable Approximation (BPDA) is a powerful adversarial attack which was used to successfully attacked several gradient obfuscated defences (Athalye et al., 2018). In this section, we show that BPDA cannot break the bias classifier, which shows that the bias classifier is essentially different from gradient obfuscated defences.

The idea of BPDA is to approximate the non-differentiable layer with a differentiable function. To attack $\mathcal{B}_{\mathcal{F}}(x)$ with BPDA, a two-layer network $\widetilde{\mathcal{F}}$ is trained with the data $(\mathcal{F}(x), \mathcal{B}_{\mathcal{F}}(x))$ and the network $\widetilde{\mathcal{F}} \circ \mathcal{F}$ is used to generate adversarial examples for $\mathcal{B}_{\mathcal{F}}(x)$. The results are given in Table 4. Since the gradients of $\mathcal{F}^{(i)}, i = 3, 4, 5, 6$ are available, it is not necessary to use BPDA and their results are not given.

Compared to Table 6, we can see that BPDA is a little stronger than the black-box attacks, as expected, because it is specifically designed for non-gradient networks. In particular, BPDA is unable to compromise the bias classifier in the same way it can bypass gradient obfuscation defenses (Athalye et al., 2018).

## 6 CONCLUSION

In this paper, we show that the bias part of a DNN can be effectively trained as a classifier. The bias classifier is shown to have universal power to approximate classification problems. Furthermore, the bias classifier can be made provably robust in certain sense. This answers the important question of how to design neural networks that provably achieve the highest possible adversarial accuracy for certain attacks. Finally, experimental results are used to show that the bias classifier is comparable to other defensive DNNs of similar sizes against major types of adversarial attack.

**Limitations and Future Research.** The biggest problem is how to design a network suitable for bias classifier, in this paper, we still use the classic network structure $VGG$ or $ResNet$, but they may not suitable for bias classifier. For further research, the estimations in Theorems 15 and 18 are not optimal, and better estimations are desirable. Moreover, how to design a more suitable network structure for bias classifier is also an interesting question.

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

# A Proof of Theorem 4

**Theorem 4.** *Let $G : \mathbb{I}^n \to [m]$ be a classification function. Then for any $\epsilon \in (0, 1/2)$, there exists a neural network $\mathcal{F} : \mathbb{I}^n \to \mathbb{R}^m$ and an open set $D \subset \mathbb{I}^n$ with volume $V(D) < \epsilon$, such that $\widehat{\mathcal{B}}_{\mathcal{F}}(x, y) = G(x)$ for $x \in \mathbb{I} \setminus D$.*

We first prove several lemmas. In this section, the notation in Theorem 4 will be used. Let $\Gamma : \mathbb{R} \to \mathbb{R}$ be the following activation function:

$$\Gamma(x) = \begin{cases} 0 & \text{if} \quad x \le 0 \\ 1 & \text{if} \quad x > 0 \end{cases}.$$

**Lemma 19** (Theorem 5 in (Cybenko, 1989))**.** *Let $G : \mathbb{I}^n \to [m]$ be a classification function. Then for any $\epsilon > 0$, there exist $N \in \mathbb{N}_+$, $W \in \mathbb{R}^{N \times n}$, $b \in \mathbb{R}^N$, $U \in \mathbb{R}^{1 \times N}$, and a set $D \subset \mathbb{I}^n$ with $V(D) < \epsilon$, such that*

$$\mathcal{G}(x) = U \cdot \Gamma(Wx + b) : \mathbb{I}^n \to \mathbb{R}$$

*and $|\mathcal{G}(x) - G(x)| < \epsilon$ for $x \in \mathbb{I}^n \setminus D$.*

The following lemma shows that there exists a DNN with one hidden layer and activation function $\Gamma$, which can simulate any classification function.

**Lemma 20.** *Let $G : \mathbb{I}^n \to [m]$ be a classification function. For any $\epsilon \in (0, 1/2)$, there exist $N \in \mathbb{N}_+$, $W \in \mathbb{R}^{N \times n}$, $b \in \mathbb{R}^N$, $U \in \mathbb{R}^{m \times N}$, and a set $D \subset \mathbb{I}^n$ with $V(D) < \epsilon$, such that*

$$\mathcal{G}(x) = U \cdot \Gamma(Wx + b) : \mathbb{I}^n \to \mathbb{R}^m \tag{19}$$

*and $\widehat{\mathcal{G}}(x) = G(x)$ for any $x \in \mathbb{I}^n / D$.*

*Proof.* For each $l \in [m]$, define a classification function $F_l : \mathbb{I}^n \to \{0, 1\}$: $F_l(x) = 1$ if $G(x) = l$, and $F_l(x) = 0$ otherwise. By Lemma 19, for all $l \in [m]$, there exist $\overline{N} \in \mathbb{N}_+$, $W_l \in \mathbb{R}^{\overline{N} \times n}$, $b_l \in \mathbb{R}^{\overline{N}}$, $U_l \in \mathbb{R}^{1 \times \overline{N}}$, and $D_l \subset \mathbb{I}^n$ with $V(D_l) < \epsilon/m$ such that

$$\mathcal{F}^{(l)}(x) = U_l \cdot \Gamma(W_l x + b_l) : \mathbb{I}^n \to \mathbb{R}$$

satisfies $|\mathcal{F}^{(l)}(x) - F_l(x)| < \epsilon$ for $x \in \mathbb{I}^n \setminus D_l$.

We now define $\mathcal{G}$ in equation 19 such that $\mathcal{G}_l(x) = \mathcal{F}^{(l)}(x)$. We just need to define $N, W, b, U$. Let $N = \overline{N}m$, $W \in \mathbb{R}^{N \times n}$, $b \in \mathbb{R}^N$, where the $l$-th row of $W$ is the $l_2$-th row of $W_{l_1}$ and the $l$-th row of $b$ is the $l_2$-th row of $b_{l_1}$, where $l = l_1 \overline{N} + l_2$, $0 \le l_2 < \overline{N}$, and $0 \le l_1 < m$. Let $U \in \mathbb{R}^{m \times N}$ be formed as follows: for $j \in [m]$, the $j$-th row of $U$ are zeros except the $(j-1)\overline{N}$-th to the $((j-1)\overline{N} + \overline{N} - 1)$-th rows, and the values of the $((j-1)\overline{N} + k)$-th place of the $j$-th row of $U$ equal to the values of the $k$-th place of $U_l$, where $k = 0, 1, \dots, \overline{N} - 1$.

Let $D = \bigcup_{i=1}^m D_l \subset \mathbb{I}^n$. It is easy to see that $V(D) < \epsilon$. Then for $x \in O_y$ and $l \ne y$, we have $\mathcal{G}_y(x) = \mathcal{F}^{(y)}(x) > F_y(x) - \epsilon = 1 - \epsilon$ and $\mathcal{G}_l(x) = \mathcal{F}^{(y)}(x) < \epsilon$. Since $\epsilon \in (0, 1/2)$, we have $\mathcal{G}_y(x) > \mathcal{G}_l(x)$ for $x \in I^n \setminus D$ and $x \in O_y$ and hance $y = \widehat{\mathcal{G}}(x)$. The lemma is proved. $\square$

**Lemma 21.** *Let $W \in \mathbb{R}^{1 \times n}$ have nonzero entries and $b \in \mathbb{R}$. For any $a > 0$, let $Z_a = \{x \in \mathbb{I}^n \,|\, |Wx + b| < a\}$. Then $V(Z_a) \le 2a\sqrt{n}^{n-1}/||W||_2$.*

*Proof.* Let $U = \{U_1, \dots, U_n\}$ be a unit orthogonal basis of $\mathbb{R}^n$ and $U_1 = \frac{W}{||W||_2}$. If $T_i = \max_{x,y \in Z_a}\{\langle x - y, U_i \rangle\}$, then we have $V(Z_a) \le \Pi_{i=1}^n T_i$.

For $i > 1$, we have $T_i \le \max_{x,y \in Z_a} ||x - y||_2 \le \sqrt{n}$. Moreover, for any $x, y \in Z_a$, we have

$$\langle x - y, U_1 \rangle = \langle x - y, W \rangle / ||W||_2 = (Wx + b - Wy - b)/||W||_2 \le 2a/||W||_2$$

which means $T_1 \le 2a/||W||_2$. Then we have

$$V(Z_a) \le \Pi_{i=1}^n T_i \le 2a\sqrt{n}^{n-1}/||W||_2.$$

The lemma is proved. $\square$

**Lemma 22.** *The bias vector $b$ in Lemma 20 can be chosen such that each entry of $b$ is not zero.*

*Proof.* By Lemma 20, there exist $N \in \mathbb{N}_+$, $W \in \mathbb{R}^{N \times n}$, $b \in \mathbb{R}^N$, $U \in \mathbb{R}^{m \times N}$, and $D_1 \subset \mathbb{I}^n$ with $V(D_1) < \epsilon/2$, such that
$$\mathcal{G}(x) = U \cdot \Gamma(Wx + b)$$
makes $\widehat{\mathcal{G}}(x) = G(x)$ for $x \in \mathbb{I}^n \setminus D_1$.

Let $\gamma = \frac{\epsilon W_m}{4N\sqrt{n}^{n-1}}$, where $W_m = ||W||_{2,\infty}$. Assume $\widehat{b} = b - I_0(|b|)\gamma$, where $I_0(x) = 1 - \text{sign}(x)$ and when $I_0$ is treated as a vector map, it acts on each vector entry, respectively. From the construction, $\widehat{b}$ does not have zero entries, because $\widehat{b}_i = b_i$ if $b_i \neq 0$, and $\widehat{b}_i = \gamma$ if $b_i = 0$, where $b_i$ and $\widehat{b}_i$ are respectively the $i$-th rows of $b$ and $\widehat{b}$.

Let $W_i$ be the $i$-th row of $W$ and $Z_i = \{z \in \mathbb{R}^n \mid |W_i z + b_i| < \gamma\}$. By Lemma 21, we have $V(Z_i \bigcap \mathbb{I}^n) < 2\gamma\sqrt{n}^{n-1}/W_m$. We write $C_n = \sqrt{n}^{n-1}/W_m$.

Let $Z = \{x \in \mathbb{R}^n \mid \Gamma(Wx + b) \neq \Gamma(Wx + \widehat{b})\}$. We will show that $Z = \cup_{i=1}^N Z_i$. If $\Gamma(Wx + b) \neq \Gamma(Wx + \widehat{b})$, then there exists an $i \in [N]$ such that $W_i x + b_i > 0$ and $W_i x + \widehat{b}_i < 0$, or $W_i x + b_i < 0$ and $W_i x + \widehat{b}_i > 0$. If $W_i x + b_i > 0$ and $W_i x + \widehat{b}_i < 0$, then $W_i x + \widehat{b}_i = W_i x + b_i - I_0(|b|)\gamma < 0$ and hence $|W_i x + b_i| \leq \gamma$. Similarly, if $W_i x + b_i < 0$ and $W_i x + \widehat{b}_i > 0$, we also have $|W_i x + b_i| \leq \gamma$, which implies $x \in Z_i$. As a consequence, $Z = \cup_{i=1}^N Z_i$.

From $Z = \cup_{i=1}^N Z_i$, we have $V(Z \bigcap \mathbb{I}^n) < 2\gamma N C_n < \epsilon/2$, since $\gamma = \frac{\epsilon}{4NC_n}$. Let $D = D_1 \bigcup (Z \bigcap \mathbb{I}^n) \subset \mathbb{I}^n$. Then $V(D) < V(D_1) + V(Z \bigcap \mathbb{I}^n) < \epsilon$.

Finally, let
$$\mathcal{G}_1(x) = U \cdot \Gamma(Wx + \widehat{b}).$$
Then, for $x \in \mathbb{I}^n \setminus D$, we have $\Gamma(Wx + b) = \Gamma(Wx + \widehat{b})$ and hence $\widehat{\mathcal{G}}_1(x) = \widehat{\mathcal{G}}(x) = G(x)$. That is, $\widehat{\mathcal{G}}_1 = G(x)$ for $x \in \mathbb{I}^n/D$ and $\mathcal{G}_1$ satisfies the conditions of the lemma. $\square$

**Lemma 23.** *Let $\mathcal{G} : \mathbb{I}^n \to \mathbb{R}^m$ be a one-hidden-layer DNN with activation function $\Gamma(x)$, and any coordinate of its bias vector is nonzero. Then there exists a DNN $\mathcal{F}$, which has the same structure as $\mathcal{G}$, except that the activation function of $\mathcal{F}$ is ReLU, such that $\mathcal{B}_{\mathcal{F}}(x) = \mathcal{G}(x)$ for all $x \in \mathbb{I}^n$.*

*Proof.* Assume $\mathcal{G}(x) = U \cdot \Gamma(Wx + b) + c$. Let $\mathcal{F}(x) = U^{\mathcal{F}} \text{ReLU}(Wx + b) + c$, where $U^{\mathcal{F}} = U \text{diag}(\frac{1}{b_i})$ and $b_i$ is the $i$-th entry of $b$. We will show that $\mathcal{F}$ satisfies the condition of the lemma. By the definition of $\Gamma$, the constant part of $\text{ReLU}(Wx + b)$ is $b \circ \Gamma(Wx + b)$, where $\circ$ is the point-wise product. So, $\mathcal{B}_{\mathcal{F}}(x) = U^{\mathcal{F}}(b \circ \Gamma(Wx + b)) + c = U \text{diag}(\frac{1}{b_i})(b \circ \Gamma(Wx + b)) + c = U\Gamma(WX + b) + c$. $\mathcal{B}_{\mathcal{F}}(x) = \mathcal{G}(x)$ and the lemma is proved. $\square$

*Proof of Theorem 4.* By Lemma 20, there exists a $D \subset \mathbb{I}^n$ with $V(D) < \epsilon$ and a network $\mathcal{G}$ with one-hidden-layer and with activation function $\Gamma(x)$, such that $\widehat{\mathcal{G}}(x) = G(x)$. By Lemma 22, all the parameters in the bias of $\mathcal{G}$ are nonzero. Then by Lemma 23, we can obtain a network $\mathcal{F}$ with ReLU as the activation function such that $\mathcal{B}_{\mathcal{F}}(x) = \mathcal{G}(x)$, and the theorem is proved. $\square$

## B    PROOFS OF SECTION 4.3

We first prove a lemma.

**Lemma 24.** *Let $x_1, x_2 \sim \mathcal{I}(-\lambda, \lambda)$ and $z = x_1 - x_2$. Then for $a \in [0, 2\lambda]$, we have $\mathbb{P}(z < a) = \mathbb{P}(z > -a) = 1 - \frac{(2\lambda - a)^2}{8\lambda^2}$, which is denoted as $T(\lambda, a) = 1 - \frac{(2\lambda - a)^2}{8\lambda^2}$. In addition, $T(\lambda, a)$ increases with $a$ and $T(\lambda, a) \in [0.5, 1]$.*

*Proof.* Let $f(z)$ be the density function of $z$. Then $f(z) = 0$, if $z \geq 2\lambda$ or $z \leq -2\lambda$; $f(z) = \frac{2\lambda + z}{4\lambda^2}$, if $0 \geq z \geq -2\lambda$; $f(z) = \frac{2\lambda - z}{4\lambda^2}$, if $0 \leq z \leq 2\lambda$. Therefore, $\mathbb{P}(z < a) = \mathbb{P}(z > -a) = 1 - \frac{(2\lambda - a)^2}{8\lambda^2}$. $\square$

We now prove Theorem 13. The key idea of the proof is to consider two cases $||W_{R,n_x} - W_{R,y}||_{-\infty} < \mu$ and $||W_{R,n_x} - W_{R,y}||_{-\infty} > \mu$, where the first case can be handled with Lemma 24 and the second case can be proved similar to Theorem 11.

**Theorem 13.** *If $||\mathcal{J}_{\mathcal{F}}||_{\infty} < \mu/2$ and $W_R \sim \mathcal{U}_{m,n}(\lambda)$, then $\mathcal{C}(\mathcal{B}_{\overline{\mathcal{F}}}, \mathcal{A}_{\mathcal{B}_{\overline{\mathcal{F}}}, \mathcal{D}_{\overline{\mathcal{F}}}^{\frac{1}{2}}}, \mathcal{U}_{m,n}(\lambda)) \leq \mathcal{C}(\mathcal{B}_{\mathcal{F}}) + \mu n/\lambda$. Furthermore, if $\lambda > \mu n/(\epsilon \mathcal{C}(\mathcal{B}_{\mathcal{F}}))$, then $\mathcal{C}(\mathcal{B}_{\overline{\mathcal{F}}}, \mathcal{A}_{\mathcal{B}_{\overline{\mathcal{F}}}, \mathcal{D}_{\overline{\mathcal{F}}}^{\frac{1}{2}}}, \mathcal{U}_{m,n}(\lambda)) \leq (1 + \epsilon)\mathcal{C}(\mathcal{B}_{\mathcal{F}})$.*

*Proof.* Similarly to equation 15, if $||W_{R,n_x} - W_{R,y}||_{-\infty} > \mu$, then we have

$$\mathcal{A}_{\mathcal{B}_{\overline{\mathcal{F}}}, \mathcal{D}_{\overline{\mathcal{F}}}^{\frac{1}{2}}}(x) = x + \rho \operatorname{sign}(W_{R,n_x} - W_{R,y}).$$

Since $V = W_{R,n_x} - W_{R,y}$ is a random vector whose values are in $[-2\lambda, 2\lambda]$, $\operatorname{sign}(V)$ is a random vector in $\{-1, 1\}^n$. By Lemma 24 and $\widehat{\mathcal{B}}_{\mathcal{F}}(x) = \widehat{\mathcal{B}}_{\overline{\mathcal{F}}}(x)$,

$$
\begin{aligned}
&\mathcal{C}(\mathcal{B}_{\overline{\mathcal{F}}}, \mathcal{A}_{\mathcal{B}_{\overline{\mathcal{F}}}, \mathcal{D}_{\overline{\mathcal{F}}}^{\frac{1}{2}}}, \mathcal{U}_{m,n}(\lambda)) \\
&= \mathbb{E}_{x\sim\mathcal{D}} \mathbb{E}_{W_R\sim\mathcal{U}_{m,n}(\lambda)}[\mathbf{I}(\widehat{\mathcal{B}}_{\mathcal{F}}(\mathcal{A}_{\mathcal{B}_{\overline{\mathcal{F}}}, \mathcal{D}_{\overline{\mathcal{F}}}^{\frac{1}{2}}}(x)) \neq \widehat{\mathcal{B}}_{\mathcal{F}}(x))] \\
&\leq \mathbb{E}_{x\sim\mathcal{D}} \mathbb{E}_{W_R\sim\mathcal{U}_{m,n}(\lambda)}[\mathbf{I}(||W_{R,n_x} - W_{R,y}||_{-\infty} \leq \mu) + \\
&\qquad \mathbf{I}(||W_{R,n_x} - W_{R,y}||_{-\infty} > \mu)[\mathbf{I}(\widehat{\mathcal{B}}_{\mathcal{F}}(\mathcal{A}_{\mathcal{B}_{\overline{\mathcal{F}}}, \mathcal{D}_{\overline{\mathcal{F}}}^{\frac{1}{2}}}(x)) \neq \widehat{\mathcal{B}}_{\mathcal{F}}(x))] \\
&\leq (1 - 2^n(1 - T(\lambda, \mu))^n) + \mathbb{E}_{x\sim\mathcal{D}} \sum_{V\in\{-1,1\}^n} \mathbf{I}(\widehat{\mathcal{B}}_{\mathcal{F}}(x + \rho V) \neq \widehat{\mathcal{B}}_{\mathcal{F}}(x))] \\
&\leq (1 - 2^n(1 - T(\lambda, \mu))^n) + \mathcal{C}(\mathcal{B}_{\mathcal{F}})
\end{aligned}
$$

where $T(\lambda, \mu) = 1 - \frac{(2\lambda - \mu)^2}{8\lambda^2}$ is introduced in Lemma 24. We have $2^n(1 - T(\lambda, \mu))^n = (2 - 2 + \frac{4\lambda^2 + \mu^2 - 4\lambda\mu}{4\lambda^2})^n = (1 - \frac{4\lambda\mu - \mu^2}{4\lambda^2})^n \geq 1 - n\frac{4\lambda\mu - \mu^2}{4\lambda^2} \geq 1 - n\mu/\lambda$. So,

$$\mathcal{C}(\mathcal{B}_{\overline{\mathcal{F}}}, \mathcal{A}_{\mathcal{B}_{\overline{\mathcal{F}}}, \mathcal{D}_{\overline{\mathcal{F}}}^{\frac{1}{2}}}, \mathcal{U}_{m,n}(\lambda)) \leq 1 - 2^n(1 - T(\lambda, \mu))^n + \mathcal{C}(\mathcal{B}_{\mathcal{F}}) \leq \mathcal{C}(\mathcal{B}_{\mathcal{F}}) + n\mu/\lambda = \mathcal{C}(\mathcal{B}_{\overline{\mathcal{F}}}) + n\mu/\lambda.$$

The theorem is proved. $\square$

## C PROOFS OF SECTION 4.4

### C.1 PROOF OF THEOREM 14

**Theorem 14.** *For $\lambda \in \mathbb{R}_+$, if $||J_{\mathcal{F}}||_{\infty} < \lambda/2$, $W_R \sim \mathcal{M}_{m,n}(\lambda)$, and $m = 2$, then $\mathcal{B}_{\overline{\mathcal{F}}}$ is information-theoretically safe against attack $\mathcal{A}_{\mathcal{B}_{\overline{\mathcal{F}}}, \mathcal{D}_{\overline{\mathcal{F}}}^2}$.*

*Proof.* Let $y \in \{0, 1\}$ be the label of $x$. Use the notation introduced in the proof of Theorem 11. Since the loss function is $L_{\text{CE}}$ and $m = 2$, we have

$$
\begin{aligned}
\frac{\nabla L_{CE}(\overline{\mathcal{F}}(x), y)}{\nabla x} &= \frac{\sum_{i=1}^m e^{\overline{\mathcal{F}}_i}(W_{x,i} - W_{x,y} + W_{R,i} - W_{R,y})}{\sum_{i=1}^m e^{\overline{\mathcal{F}}_i(x)}} \\
&= \frac{e^{\overline{\mathcal{F}}_{1-y}(x)}}{\sum_{i=1}^m e^{\overline{\mathcal{F}}_i(x)}}(W_{x,1-y} - W_{x,y} + W_{R,1-y} - W_{R,y}).
\end{aligned}
\tag{20}
$$

The last equality comes from $m = 2$. Since $||W_{x,i} - W_{x,j}||_{\infty} < \lambda$ and $||W_{R,i} - W_{R,j}||_{-\infty} > \lambda$ for $i \neq j$, we have $\mathcal{D}_{\overline{\mathcal{F}}}^2(x, y) = \operatorname{sign}(\frac{\nabla L_{CE}(\overline{\mathcal{F}}(x), y)}{\nabla x}) = \operatorname{sign}(W_{R,1-y} - W_{R,y})$ which is a random vector in $\{-1, 1\}^n$, similar to the proof of Theorem 11. The theorem is proved. $\square$

### C.2 PROOF OF THEOREM 15

We first prove two lemmas.

**Lemma 25.** *Let $\{u_i\}_{i=1}^n$ be a set of iid random variables with values in $[-\lambda, \lambda]$ and $u = \sum_{i=1}^n x_i u_i$, where $x_i \in \mathbb{R}$ such that $|x_i| > a > 0$ for some $i$. Let the density function of $u$ be $f(x)$. Then $f(x) < \frac{1}{2\lambda a}$ for all $x$.*

*Proof.* Assume $|x_n| > a$ and $f_n(x)$ is the distribution function of $x_n u_n$. We have

$$\mathbb{P}(u < m) = \int_{\{-\lambda|x_i|\}_{i=1}^{n-1}}^{\{\lambda|x_i|\}_{i=1}^{n-1}} (\Pi_{i=1}^{n-1} \frac{1}{2\lambda|x_i|}) f_n(m - \sum_{i=1}^{n-1} t_i) \mathrm{d}t_1 t_2 \dots t_{n-1}.$$

Since $0 < f_n'(x) \leq \frac{1}{2\lambda|x_n|}$ and $f(x) = \frac{\nabla \mathbb{P}(u<x)}{\nabla x}$, we have

$$
\begin{aligned}
&f(x) \\
&= \frac{\nabla \mathbb{P}(u<x)}{\nabla x} \\
&= \frac{\nabla \int_{\{-\lambda|x_i|\}_{i=1}^{n-1}}^{\{\lambda|x_i|\}_{i=1}^{n-1}} (\Pi_{i=1}^{n-1} \frac{1}{2\lambda|x_i|}) f_n(x - \sum_{i=1}^{n-1} t_i) \mathrm{d}t_1 t_2 \dots t_{n-1}}{\nabla x} \\
&= \int_{\{-\lambda|x_i|\}_{i=1}^{n-1}}^{\{\lambda|x_i|\}_{i=1}^{n-1}} (\Pi_{i=1}^{n-1} \frac{1}{2\lambda|x_i|}) \frac{\nabla f_n(x - \sum_{i=1}^{n-1} t_i)}{\nabla x} \mathrm{d}t_1 t_2 \dots t_{n-1} \\
&\leq \int_{\{-\lambda|x_i|\}_{i=1}^{n-1}}^{\{\lambda|x_i|\}_{i=1}^{n-1}} (\Pi_{i=1}^{n-1} \frac{1}{2\lambda|x_i|}) \frac{1}{2\lambda|x_n|} \mathrm{d}t_1 t_2 \dots t_{n-1} \\
&\leq \frac{1}{2\lambda|x_n|} \\
&\leq \frac{1}{2\lambda a}.
\end{aligned}
$$

The lemma is proved. $\qquad\square$

**Lemma 26.** *Let $\{u_i\}_{i=1}^n$ be a set of iid variables, $f_i$ the density function of $u_i$, and $f_i(x) < a$ for all $x \in \mathbb{R}$. Then we have*

$$\mathbb{P}(|u_i - u_j| > \psi \text{ for } \forall i \neq j) > \Pi_{i=0}^{n-1}(1 - 2i\psi a).$$

*Proof.* Let $D_k$ be the event $|u_i - u_j| > \psi$ for $\forall i, j \leq k$, and $F_k : \mathbb{R}^k \to \mathbb{R}$ the joint probability density function of $\{u_i\}_{i=1}^k$ under condition $D_k$. Then we have

$$
\begin{aligned}
&\mathbb{P}(D_k) \\
=\ & \mathbb{P}(D_k, D_{k-1}) \\
=\ & \mathbb{P}(D_k \| D_{k-1}) \mathbb{P}(D_{k-1}) \\
=\ & \mathbb{P}(|u_k - u_i| > \psi \text{ for } \forall i < k \| D_{k-1}) \mathbb{P}(D_{k-1}) \\
=\ & \mathbb{P}(D_{k-1}) \int_{-\infty^{k-1}}^{\infty} \int_{-\infty}^{\infty} F_{k-1}(t_1, \dots, t_{k-1}) f_k(t_k) I(|t_k - t_i| > \psi \forall i < k) \mathrm{d}t_k \mathrm{d}t_1 \dots t_{k-1} \\
>\ & \mathbb{P}(D_{k-1}) \int_{-\infty^{k-1}}^{\infty} \int_{-\infty}^{\infty} F_{k-1}(t_1, \dots, t_{k-1})(f_k(t_k) - aI(|t_k - t_i| < \psi \exists i < k)) \mathrm{d}t_k \mathrm{d}t_1 \dots t_{k-1} \\
=\ & \mathbb{P}(D_{k-1})(1 - \int_{-\infty^{k-1}}^{\infty} \int_{-\infty}^{\infty} a F_{k-1}(t_1, \dots, t_{k-1}) I(|t_k - t_i| < \psi \exists i < k) \mathrm{d}t_k \mathrm{d}t_1 \dots t_{k-1}) \\
>\ & \mathbb{P}(D_{k-1})(1 - \int_{-\infty^{k-1}}^{\infty} 2a(k-1)\psi F_{k-1}(t_1, \dots, t_{k-1}) \mathrm{d}t_1 \dots t_{k-1}) \\
=\ & \mathbb{P}(D_{k-1})(1 - 2a(k-1)\psi).
\end{aligned}
$$

Since $\mathbb{P}(D_0) = 1$, we have

$$
\begin{aligned}
&\mathbb{P}(|u_i - u_j| > \psi \text{ for } \forall i \neq j) \\
=\ & \mathbb{P}(D_n) \\
>\ & \mathbb{P}(D_{n-1})(1 - 2(n-1)\psi a) \\
>\ & \mathbb{P}(D_{n-2})(1 - 2(n-1)\psi a)(1 - 2(n-2)\psi a) \\
>\ & \dots \\
>\ & \Pi_{i=0}^{n-1}(1 - 2i\psi a).
\end{aligned}
$$

The lemma is proved. $\qquad\square$

**Theorem 15.** *Let* $||J_{\mathcal{F}}||_\infty < \mu/2$, $||\mathcal{B}_{\mathcal{F}}||_\infty < \beta$, *and* $\lambda \in \mathbb{R}_+$ *satisfy* $\lambda > \mu$ *and* $(\lambda - \mu)e^{-2\beta - n\mu + \sqrt{\lambda}} > (2m\lambda + \mu)m$. *Furthermore, assume that the samples are normalized, that is,* $||x||_\infty = 1$. *If* $W_R \sim \mathcal{M}_{m,n}(\lambda)$, *then* $\mathcal{C}(\mathcal{B}_{\overline{\mathcal{F}}}, \mathcal{A}_{\mathcal{B}_{\overline{\mathcal{F}}}, \mathcal{D}_{\overline{\mathcal{F}}}^2}, \mathcal{M}_{m,n}(\lambda)) \le (m-1)\mathcal{C}(\mathcal{B}_{\mathcal{F}}) + \frac{(m-2)^2}{\sqrt{\lambda}}$.

*Proof.* From equation 2 and equation 12, we have $\mathcal{F}(x) = W_x x + B_x$ and $\overline{\mathcal{F}}(x) = (W_x + W_R)x + B_x$. Let $x$ be a sample with label $y$. From equation equation 20, we have

$$\frac{\nabla L_{CE}(\overline{\mathcal{F}}(x), y)}{\nabla x} = \frac{\sum_{i=1}^m (W_{R,i} - W_{R,y} + W_{x,i} - W_{x,y})e^{\overline{\mathcal{F}}_i(x)}}{\sum_{i=1}^m e^{\overline{\mathcal{F}}_i(x)}}.$$

Let $m_x = \arg\max_{i \ne y}\{\langle W_{R,i}, x\rangle\}$ and consider the condition:

**Condition $C_1$:** $\langle W_{R,m_x}, x\rangle > \langle W_{R,j}, x\rangle + \sqrt{\lambda}$ for all $j \in [m] \setminus \{y, m_x\}$.

We first give the probability for condition $C_1$ to be valid. By Lemmas 25 and 26 and due to $|x|_\infty = 1$, we have

$$
\begin{aligned}
&\mathbb{E}_{x \sim \mathcal{D}}\mathbb{P}_{W_R \sim \mathcal{M}_{m,n}(\lambda)}(C_1) \\
\ge\ & \mathbb{E}_{x \sim \mathcal{D}}\mathbb{P}_{W_R \sim \mathcal{M}_{m,n}(\lambda)}(|\langle W_{R,j}, x\rangle - \langle W_{R,i}, x\rangle| > \sqrt{\lambda}, \forall i, j \in [m]/\{y\}, \ i \ne j) \\
\ge\ & \mathbb{E}_{x \sim \mathcal{D}}\Pi_{i=1}^{m-2}(1 - \frac{2i\sqrt{\lambda}}{2\lambda|x|_\infty}) \\
\ge\ & (1 - \frac{m-2}{\sqrt{\lambda}})^{m-2} \\
\ge\ & 1 - \frac{(m-2)^2}{\sqrt{\lambda}}.
\end{aligned}
\tag{21}
$$

Let $||x||_{-\infty} = \min_{i \in [n]}\{|x|_i\}$ for $x \in \mathbb{R}^n$. Since $|\frac{\nabla\mathcal{F}(x)}{\nabla x}|_\infty < \mu/2$ and $W_R \sim \mathcal{M}_{m,n}(\lambda)$, we have $||W_{R,i} + W_{R,j}||_{-\infty} > \lambda$, $||W_{R,i} + W_{R,j}||_\infty < 2m\lambda$ and $||W_{x,i} + W_{x,j}||_\infty < \mu$ for any $i \ne j$. If condition $C_1$ is satisfied, then for any $j \in [m] \setminus \{y, m_x\}$, we have

$$
\begin{aligned}
&\overline{\mathcal{F}}_{m_x}(x) - \overline{\mathcal{F}}_j(x) \\
=\ & (W_{R,m_x} + W_{x,m_x} - W_{R,j} - W_{x,j})x + \mathcal{B}_{x,m_x} - \mathcal{B}_{x,j} \\
=\ & (W_{R,m_x} - W_{R,j})x + (W_{x,m_x} - W_{x,j})x + \mathcal{B}_{x,m_x} - \mathcal{B}_{x,j} \\
>\ & \sqrt{\lambda} - n\mu - 2\beta.
\end{aligned}
$$

Further considering the hypothesis $(\lambda - \mu)e^{-2\beta - n\mu + \sqrt{\lambda}} > (2m\lambda + \mu)m$, we have

$$
\begin{aligned}
&||W_{R,m_x} - W_{R,y} + W_{x,m_x} - W_{x,y}||_{-\infty}e^{\overline{\mathcal{F}}_{m_x}(x)} \\
>\ & (\lambda - \mu)e^{\overline{\mathcal{F}}_{m_x}(x)} \\
>\ & (\lambda - \mu)e^{\overline{\mathcal{F}}_j(x) + \sqrt{\lambda} - 2\beta - n\mu} \\
=\ & (\lambda - \mu)e^{-2\beta - n\mu}e^{\sqrt{\lambda}}e^{\overline{\mathcal{F}}_j(x)} \\
>\ & (2m\lambda + \mu)me^{\overline{\mathcal{F}}_j(x)} \\
>\ & m||(W_{R,j} - W_{R,y} + W_{x,j} - W_{x,y})||_\infty e^{\overline{\mathcal{F}}_j(x)}
\end{aligned}
$$

which means

$$\text{sign}(\sum_{i=1}^m (W_{R,i} - W_{R,y} + W_{x,i} - W_{x,y})e^{\overline{\mathcal{F}}_i(x)}) = \text{sign}((W_{R,m_x} - W_{R,y} + W_{x,m_x} - W_{x,y})e^{\overline{\mathcal{F}}_{m_x}(x)}).$$

Because of this, we have

$$
\begin{aligned}
&\text{sign}(\frac{\nabla L(\overline{\mathcal{F}}(x), y)}{\nabla x}) \\
=\ & \text{sign}(\frac{\sum_{i=1}^m (W_{R,i} - W_{R,y} + W_{x,i} - W_{x,y})e^{\overline{\mathcal{F}}_i(x)}}{\sum_{i=1}^m e^{\overline{\mathcal{F}}_i(x)}}) \\
=\ & \text{sign}(\sum_{i=1}^m (W_{R,i} - W_{R,y} + W_{x,i} - W_{x,y})e^{\overline{\mathcal{F}}_i(x)}) \\
=\ & \text{sign}((W_{R,m_x} - W_{R,y} + W_{x,m_x} - W_{x,y})e^{\overline{\mathcal{F}}_{m_x}(x)}) \\
=\ & \text{sign}((W_{R,m_x} - W_{R,y} + W_{x,m_x} - W_{x,y}) \\
=\ & \text{sign}(W_{R,m_x} - W_{R,y}).
\end{aligned}
$$

Let $V$ be a random vector in $\{0,1\}^n$. Then the probability for the sign of $W_{R,m_x} - W_{R,y}$ to be $V$ is

$$\mathbb{P}(\text{sign}(W_{R,m_x} - W_{R,y}) = V, C_1)$$
$$\leq \quad \mathbb{P}(\text{sign}(W_{R,m_x} - W_{R,y}) = V)$$
$$= \quad \sum_{i<y}\mathbb{P}(m_x = i, \text{sign}(W_{R,y}) = V) + \sum_{i>y}\mathbb{P}(m_x = i, \text{sign}(W_{R,i}) = V)$$
$$\leq \quad \sum_{i<y}\mathbb{P}(\text{sign}(W_{R,y}) = V) + \sum_{i>y}\mathbb{P}(\text{sign}(W_{R,i}) = V)$$
$$= \quad \frac{m-1}{2^n}.$$

So we have

$$\mathbb{E}_{W_R \sim \mathcal{M}_{m,n}(\lambda)}[\mathbf{I}(\widehat{\mathcal{B}}_{\mathcal{F}}(x + \rho\text{sign}(\tfrac{\nabla L_{CE}(\overline{\mathcal{F}}(x),y)}{\nabla x})) \neq \widehat{\mathcal{B}}_{\mathcal{F}}(x))\mathbf{I}(C_1)]$$
$$= \quad \mathbb{E}_{W_R \sim \mathcal{M}_{m,n}(\lambda)}[\mathbf{I}(\widehat{\mathcal{B}}_{\mathcal{F}}(x + \rho\text{sign}(W_{R,m_x} - W_{R,y})) \neq \widehat{\mathcal{B}}_{\mathcal{F}}(x))\mathbf{I}(C_1)]$$
$$= \quad \sum_{V \in \{-1,1\}^n}\mathbb{P}(\text{sign}(W_{R,m_x} - W_{R,y}) = V,\ C_1)\mathbf{I}(\widehat{\mathcal{B}}_{\mathcal{F}}(x + \rho V) \neq \widehat{\mathcal{B}}_{\mathcal{F}}(x))$$
$$\leq \quad \sum_{V \in \{-1,1\}^n}(m-1)/(2^n)\mathbf{I}(\widehat{\mathcal{B}}_{\mathcal{F}}(x + \rho V) \neq \widehat{\mathcal{B}}_{\mathcal{F}}(x))$$
$$= \quad (m-1)\mathcal{C}(\mathcal{B}_{\mathcal{F}}).$$

Finally, from equation 21 we have

$$\mathcal{C}(\mathcal{B}_{\overline{\mathcal{F}}}, \mathcal{A}_{\mathcal{B}_{\overline{\mathcal{F}}}, \mathcal{D}^2_{\overline{\mathcal{F}}}}, \mathcal{M}_{m,n}(\lambda))$$
$$= \quad \mathbb{E}_{W_R \sim \mathcal{M}_{m,n}(\lambda)}\mathbb{E}_{x \sim \mathcal{D}}[\mathbf{I}(\widehat{\mathcal{B}}_{\mathcal{F}}(x + \rho\text{sign}(\tfrac{\nabla L(\overline{\mathcal{F}}(x),y)}{\nabla x})) \neq \widehat{\mathcal{B}}_{\mathcal{F}}(x))]$$
$$\leq \quad \mathbb{E}_{x \sim \mathcal{D}}\mathbb{E}_{W_R \sim \mathcal{M}_{m,n}(\lambda)}[\mathbf{I}(\widehat{\mathcal{B}}_{\mathcal{F}}(x + \rho\text{sign}(\tfrac{\nabla L(\overline{\mathcal{F}}(x),y)}{\nabla x})) \neq \widehat{\mathcal{B}}_{\mathcal{F}}(x))\mathbf{I}(C_1) + (1 - \mathbf{I}(C_1))]$$
$$\leq \quad (m-1)\mathcal{C}(\mathcal{B}_{\mathcal{F}}) + \mathbb{E}_{x \sim \mathcal{D}}\mathbb{E}_{W_R \sim \mathcal{M}_{m,n}(\lambda)}[(1 - \mathbf{I}(C_1))]$$
$$\leq \quad (m-1)\mathcal{C}(\mathcal{B}_{\mathcal{F}}) + \mathbb{E}_{x \sim \mathcal{D}}[1 - \mathbb{P}_{W_R \sim \mathcal{M}_{m,n}(\lambda)}(C_1)]$$
$$\leq \quad (m-1)\mathcal{C}(\mathcal{B}_{\mathcal{F}}) + \frac{(m-2)^2}{\sqrt{\lambda}}.$$

The theorem is proved. $\qquad\square$

## C.3 PROOF OF THEOREM 17

**Theorem 17.** *If* $||\mathcal{J}_{\mathcal{F}}||_\infty < \mu/2$, $W_R \sim \mathcal{U}_{m,n}(\lambda)$, *and* $m = 2$, *then* $\mathcal{C}(\mathcal{B}_{\overline{\mathcal{F}}}, \mathcal{A}_{\mathcal{B}_{\overline{\mathcal{F}}}, \mathcal{D}^2_{\overline{\mathcal{F}}}}, \mathcal{U}_{m,n}(\lambda)) \leq$
$e^{n\mu/\lambda}\mathcal{C}(\mathcal{B}_{\mathcal{F}})$. *Furthermore, if* $\lambda > n\mu/\ln(1 + \epsilon)$, *then* $\mathcal{C}(\mathcal{B}_{\overline{\mathcal{F}}}, \mathcal{A}_{\mathcal{B}_{\overline{\mathcal{F}}}, \mathcal{D}^2_{\overline{\mathcal{F}}}}, \mathcal{U}_{m,n}(\lambda)) \leq (1 + \epsilon)\mathcal{C}(\mathcal{B}_{\mathcal{F}})$.

*Proof.* Let $y \in \{0,1\}$ be the label of $x$. Denote $U = W_{x,1-y} - W_{x,y} \in \mathbb{R}^{1 \times n}$ and $Z = W_{R,1-y} - W_{R,y} \in \mathbb{R}^{1 \times n}$. We have

$$\text{sign}(\tfrac{\nabla L_{CE}(\mathcal{F}(x),y)}{\nabla x})$$
$$= \quad \text{sign}(\tfrac{e^{\mathcal{F}_{1-y}(x)}(\tfrac{\nabla(\mathcal{F}_{1-y}(x))}{\nabla x} - \tfrac{\nabla(\mathcal{F}_y(x))}{\nabla x})}{e^{\mathcal{F}_y(x)} + e^{\mathcal{F}_{1-y}(x)}})$$
$$= \quad \text{sign}(\tfrac{\nabla(\mathcal{F}_{1-y}(x))}{\nabla x} - \tfrac{\nabla(\mathcal{F}_y(x))}{\nabla x})$$
$$= \quad \text{sign}(W_{x,1-y} - W_{x,y})$$
$$= \quad \text{sign}(U).$$

From equation equation 20, we have

$$\text{sign}(\tfrac{\nabla L_{CE}(\overline{\mathcal{F}}(x),y)}{\nabla x})$$
$$= \quad \text{sign}(\tfrac{e^{\overline{\mathcal{F}}_{1-y}(x)}}{\sum_{i=1}^m e^{\overline{\mathcal{F}}_i(x)}}(W_{x,1-y} - W_{x,y} + W_{R,1-y} - W_{R,y})).$$
$$= \quad \text{sign}(U + Z).$$

For $i \in [n]$, $\text{sign}(U_i) = \text{sign}(U_i + Z_i)$ if and only if ($Z_i \leq -U_i$ when $U_i \leq 0$) or ($Z_i \geq -U_i$ when $U_i \geq 0$), where $Z_i, U_i$ are respectively the $i$-th coordinates of $Z, U$. Since $W_R \sim \mathcal{U}_{m,n}(\lambda)$,

$Z = W_{R,1-y} - W_{R,y}$ is the difference of two uniform distributions in $[-\lambda, \lambda]$. By Lemma 24, $U_i > 0$ implies $\mathbb{P}(Z_i \geq -U_i) = T(\lambda, |U_i|) < T(\lambda, \mu)$, and $U_i < 0$ implies $\mathbb{P}(Z_i \leq -U_i) = T(\lambda, |U_i|) < T(\lambda, \mu)$. Hence, no matter what is the value of $U$, we always have $\mathbb{P}(\text{sign}(U) = \text{sign}(U + Z)) < T(\lambda, \mu)^n$, where $T(\lambda, \mu) = 1 - \frac{(2\lambda - \mu)^2}{8\lambda^2}$.

Moreover, for $i \in [n]$, if $\text{sign}(U_i) \neq \text{sign}(U_i + Z_i)$, we have ($Z_i > 0$ when $U_i < 0$) or ($Z_i < 0$ when $U_i > 0$). So, $\mathbb{P}(\text{sign}(U_i) \neq \text{sign}(U_i + Z_i)) < 1/2 < T(\lambda, \mu)$, since $T(\lambda, \mu)$ is always $\geq 1/2$.

Since $\{Z_i\}_{i \in [n]}$ is iid, by Lemma 24, for a random vector $V \in \{-1, 1\}^n$ we have

$$\mathbb{P}_{W_R \sim \mathcal{U}_{m,n}(\lambda)}(\text{sign}(\frac{\nabla L_{CE}(\overline{\mathcal{F}}(x), y)}{\nabla x}) = V)$$
$$= \mathbb{P}_{W_R \sim \mathcal{U}_{m,n}(\lambda)}(\text{sign}(U + Z) = V)$$
$$= \prod_{i=1}^{n} \mathbb{P}_{W_R \sim \mathcal{U}_{m,n}(\lambda)}(\text{sign}(U_i + Z_i) = V_i)$$
$$= \prod_{i=1}^{n} (\mathbf{I}(\text{sign}(U_i) = V_i)\mathbb{P}_{W_R \sim \mathcal{U}_{m,n}(\lambda)}(\text{sign}(U_i) = \text{sign}(U_i + Z_i))$$
$$+ \mathbf{I}(\text{sign}(U_i) \neq V_i)\mathbb{P}_{W_R \sim \mathcal{U}_{m,n}(\lambda)}(\text{sign}(U_i) \neq \text{sign}(U_i + Z_i)))$$
$$\leq \prod_{i=1}^{n} (\mathbf{I}(\text{sign}(U_i) = V_i)T(\lambda, \mu) + \mathbf{I}(\text{sign}(U_i) \neq V_i)T(\lambda, \mu))$$
$$= T(\lambda, \mu)^n.$$

For $V \in \{-1, 1\}^n$, denote $Q(x, V) = \mathbf{I}(\widehat{\mathcal{B}}_{\mathcal{F}}(x + \rho V) \neq \widehat{\mathcal{B}}_{\mathcal{F}}(x))$. We have

$$\mathcal{C}(\mathcal{B}_{\overline{\mathcal{F}}}, \mathcal{A}_{\mathcal{B}_{\overline{\mathcal{F}}}, \mathcal{D}_{\overline{\mathcal{F}}}^2}, \mathcal{U}_{m,n}(\lambda))$$
$$= \mathbb{E}_{x \sim D_0}\mathbb{E}_{W_R \sim \mathcal{U}_{m,n}(\lambda)}[\mathbf{I}(\widehat{\mathcal{B}}_{\mathcal{F}}(x + \rho\text{sign}(\frac{\nabla L_{CE}(\overline{\mathcal{F}}(x), y)}{\nabla x})) \neq \widehat{\mathcal{B}}_{\mathcal{F}}(x))]]$$
$$= \mathbb{E}_{x \sim D_0}[\sum_{V \in \{-1,1\}^n} \mathbb{P}_{W_R \sim \mathcal{U}_{m,n}(\lambda)}(\text{sign}(\frac{\nabla L_{CE}\overline{\mathcal{F}}(x), y)}{\nabla x}) = V)Q(x, V)]$$
$$\leq (T(\lambda, \mu))^n\mathbb{E}_{x \sim D_0}[(\sum_{V \in \{-1,1\}^n} Q(x, V))]$$
$$\leq (2T(\lambda, \mu))^n\mathcal{C}(\mathcal{B}_{\mathcal{F}})$$

where $T(\lambda, \mu) = 1 - \frac{(2\lambda - \mu)^2}{8\lambda^2}$. We have $(2T(\lambda, \mu))^n = (2 - \frac{4\lambda^2 + \mu^2 - 4\lambda\mu}{4\lambda^2})^n = (1 + \frac{4\lambda\mu - \mu^2}{4\lambda^2})^n \leq (1 + \frac{\mu}{\lambda})^n \leq e^{n\mu/\lambda}$. Therefore, $\mathcal{C}(\mathcal{B}_{\overline{\mathcal{F}}}, \mathcal{A}_{\mathcal{B}_{\overline{\mathcal{F}}}, \mathcal{D}_{\overline{\mathcal{F}}}^2}, \mathcal{U}_{m,n}(\lambda)) < e^{n\mu/\lambda}\mathcal{C}(\mathcal{B}_{\mathcal{F}})$. The theorem is proved. $\qquad\square$

## C.4 PROOF OF THEOREM 18

**Theorem 18.** *Assume* $||\mathcal{J}_{\mathcal{F}}||_\infty < \mu/4$, $||\mathcal{B}_{\mathcal{F}}||_\infty < \beta$, *and* $\lambda \in \mathbb{R}_+$ *satisfying* $\mu e^{-2\beta - n\mu/2 + \sqrt{\lambda}} > 2(2\lambda + \mu)m$. *Furthermore, assume that the samples are normalized, that is,* $||x||_\infty = 1$. *If* $W_R \sim \mathcal{U}_{m,n}(\lambda)$, *then* $\mathcal{C}(\mathcal{B}_{\overline{\mathcal{F}}}, \mathcal{A}_{\mathcal{B}_{\overline{\mathcal{F}}}, \mathcal{D}_{\overline{\mathcal{F}}}^2}, \mathcal{U}_{m,n}(\lambda)) \leq (m-1)\mathcal{C}(\mathcal{B}_{\mathcal{F}}) + \frac{(m-1)n\mu}{\lambda} + \frac{(m-2)^2}{\sqrt{\lambda}}$.

*Proof.* The proof is similar to that of Theorem 15. So certain details of the proof are omitted. From equation equation 20, we have

$$\frac{\nabla L_{CE}(\overline{\mathcal{F}}(x), y)}{\nabla x} = \frac{\sum_{i=1}^{m}(W_{R,i} + W_{x,i} - W_{R,y} - W_{x,y})e^{\overline{\mathcal{F}}_i(x)}}{\sum_{i=1}^{m} e^{\overline{\mathcal{F}}_i(x)}}.$$

Let $m_x = \arg\max_{i \neq y}\{\langle W_{R,i}, x \rangle\}$ and consider two conditions $C_1$ and $C_2$:

**Conditions** $C_1$: $\langle W_{R,m_x}, x \rangle > \langle W_{R,j}, x \rangle + \sqrt{\lambda}$ for all $j \in [m] \setminus \{y, m_x\}$.

**Conditions** $C_2$: $||W_{R,m_x} - W_{R,y}||_{-\infty} > \mu$.

Note that condition $C_2$ implies $\text{sign}((W_{R,i} - W_{R,y} + W_{x,i} - W_{x,y}) = \text{sign}(W_{R,i} - W_{R,y})$.

We give the probabilities for conditions $C_1$ and $C_2$ to be valid. From the proof of Theorem 15,

$$\mathbb{E}_{x \sim \mathcal{D}}\mathbb{P}_{W_R \sim \mathcal{M}_{m,n}(\lambda)}(C_1) \geq 1 - \frac{(m-2)^2}{\sqrt{\lambda}}.$$

Let $f(x)$ be the density function of $W_{R,m_x}$. Then

$$
\begin{aligned}
&\mathbb{P}_{W_R \sim \mathcal{U}_{m,n}(\lambda)}(C_2) \\
&\geq \mathbb{P}_{W_R \sim \mathcal{U}_{m,n}(\lambda)}(||W_{R,i} - W_{R,y}||_{-\infty} > \mu, \forall i \neq y) \\
&\geq (1 - \tfrac{(m-1)\mu}{\lambda})^n \\
&\geq 1 - \tfrac{(m-1)n\mu}{\lambda}.
\end{aligned}
$$

For $V \in \{-1, 1\}^n$, it is also easy to see

$$
\begin{aligned}
&\mathbb{P}(\text{sign}(W_{R,m_x} - W_{R,y}) = V,\ C_1,\ C_2) \\
&\leq\ \mathbb{P}(\text{sign}(W_{R,m_x} - W_{R,y}) = V) \\
&=\ \sum_{i<y} \mathbb{P}(m_x = i, \text{sign}(W_{R,y}) = V) + \sum_{i>y} \mathbb{P}(m_x = i, \text{sign}(W_{R,i}) = V) \\
&\leq\ \sum_{i<y} \mathbb{P}(\text{sign}(W_{R,y}) = V) + \sum_{i>y} \mathbb{P}(\text{sign}(W_{R,i}) = V) \\
&=\ \tfrac{m-1}{2^n}.
\end{aligned}
$$

If conditions $C_1$ and $C_2$ are satisfied, then for any $y \in [m] \setminus \{y, m_x\}$, we have

$$
\begin{aligned}
&||W_{R,m_x} + W_{x,m_x} - W_{R,y} - W_{x,y}||_{-\infty} e^{\overline{\mathcal{F}}_{m_x}(x)} \\
&>\ \mu/2 e^{\overline{\mathcal{F}}_{m_x}(x)} \\
&>\ \mu/2 e^{\overline{\mathcal{F}}_j(x) + \sqrt{\lambda} - 2\beta - n\mu/2} \\
&=\ \mu/2 e^{-2b - n\mu/2} e^{\sqrt{\lambda}} e^{\overline{\mathcal{F}}_j(x)} \\
&>\ (2\lambda + \mu)m e^{\overline{\mathcal{F}}_j(x)} \\
&>\ m||W_{R,j} + W_{x,j} - W_{R,y} - W_{x,y}||_{\infty} e^{\overline{\mathcal{F}}_j(x)}
\end{aligned}
$$

which means

$$
\begin{aligned}
&\text{sign}(\sum_{i=1}^m (W_{R,i} + W_{x,i} - W_{R,y} - W_{x,y}) e^{\overline{\mathcal{F}}_i(x)}) \\
&=\ \text{sign}((W_{R,m_x} + W_{x,m_x} - W_{R,y} - W_{x,y}) e^{\overline{\mathcal{F}}_{m_x}(x)}),
\end{aligned}
$$

and hence

$$
\begin{aligned}
&\text{sign}(\tfrac{\nabla L_{CE}(\overline{\mathcal{F}}(x), y)}{\nabla x}) \\
&=\ \text{sign}(\tfrac{\sum_{i=1}^m (W_{R,i} + W_{x,i} - W_{R,y} - W_{x,y}) e^{\overline{\mathcal{F}}_i(x)}}{\sum_{i=1}^m e^{\overline{\mathcal{F}}_i(x)}}) \\
&=\ \text{sign}(\sum_{i=1}^m (W_{R,i} + W_{x,i} - W_{R,y} - W_{x,y}) e^{\overline{\mathcal{F}}_i(x)}) \\
&=\ \text{sign}((W_{R,m_x} + W_{x,m_x} - W_{R,y} - W_{x,y}) e^{\overline{\mathcal{F}}_{m_x}(x)}) \\
&=\ \text{sign}(W_{R,m_x} + W_{x,m_x} - W_{R,y} - W_{x,y}) \\
&=\ \text{sign}(W_{R,m_x} - W_{R,y}).
\end{aligned}
$$

Hence

$$
\begin{aligned}
&\mathbb{E}_{W_R \sim \mathcal{U}_{m,n}(\lambda)}[\mathbf{I}(\widehat{\mathcal{B}}_{\mathcal{F}}(x + \rho\,\text{sign}(\tfrac{\nabla L_{CE}(\overline{\mathcal{F}}(x), y)}{\nabla x})) \neq \widehat{\mathcal{B}}_{\mathcal{F}}(x))\mathbf{I}(C_1,\ C_2)] \\
&=\ \mathbb{E}_{W_R \sim \mathcal{U}_{m,n}(\lambda)}[\mathbf{I}(\widehat{\mathcal{B}}_{\mathcal{F}}(x + \rho\,\text{sign}(W_{R,m_x} - W_{R,y})) \neq \widehat{\mathcal{B}}_{\mathcal{F}}(x))\mathbf{I}(C_1,\ C_2)] \\
&=\ \sum_{V \in \{-1,1\}^n} \mathbb{P}(\text{sign}(W_{R,m_x} - W_{R,y}) = V,\ C_1,\ C_2)\mathbf{I}(\widehat{\mathcal{B}}_{\mathcal{F}}(x + \rho V) \neq \widehat{\mathcal{B}}_{\mathcal{F}}(x)) \\
&\leq\ \tfrac{m-1}{2^n} \sum_{V \in \{-1,1\}^n} \mathbf{I}(\widehat{\mathcal{B}}_{\mathcal{F}}(x + \rho V) \neq \widehat{\mathcal{B}}_{\mathcal{F}}(x)) \\
&=\ (m-1)\mathcal{C}(\mathcal{B}_{\mathcal{F}}).
\end{aligned}
$$

Finally, we have

$$
\begin{aligned}
&\mathcal{C}(\mathcal{B}_{\overline{\mathcal{F}}}, \mathcal{A}_{\mathcal{B}_{\overline{\mathcal{F}}}, \mathcal{D}_{\overline{\mathcal{F}}}^2}, \mathcal{U}_{m,n}(\lambda)) \\
=\ & \mathbb{E}_{x \sim D_0} \mathbb{E}_{W_R \sim \mathcal{U}_{m,n}(\lambda)}[\mathbf{I}(\widehat{\mathcal{B}}_{\mathcal{F}}(x + \rho\,\mathrm{sign}(\tfrac{\nabla L_{CE}(\overline{\mathcal{F}}(x), y)}{\nabla x})) \neq \widehat{\mathcal{B}}_{\mathcal{F}}(x))] \\
\leq\ & \mathbb{E}_{x \sim D_0} \mathbb{E}_{W_R \sim \mathcal{U}_{m,n}(\lambda)}[\mathbf{I}(\widehat{\mathcal{B}}_{\mathcal{F}}(x + \rho\,\mathrm{sign}(\tfrac{\nabla L_{CE}(\overline{\mathcal{F}}(x), y)}{\nabla x})) \neq \widehat{\mathcal{B}}_{\mathcal{F}}(x)) \\
& \mathbf{I}(C_1,\ C_2) + (1 - \mathbf{I}(C_1)) + (1 - \mathbf{I}(C_2))] \\
\leq\ & (m-1)\mathcal{C}(\mathcal{B}_{\mathcal{F}}) + \mathbb{E}_{x \sim D_0} \mathbb{E}_{W_R \sim \mathcal{U}_{m,n}(\lambda)}[(1 - \mathbf{I}(C_1)) + (1 - \mathbf{I}(C_2))] \\
\leq\ & (m-1)\mathcal{C}(\mathcal{B}_{\mathcal{F}}) + \mathbb{E}_{x \sim D_0}[1 - \mathbb{P}_{W_R \sim \mathcal{U}_{m,n}(\lambda)}(C_1)] + \mathbb{E}_{x \sim D_0}[1 - \mathbb{P}_{W_R \sim \mathcal{U}_{m,n}(\lambda)}(C_2)] \\
\leq\ & (m-1)\mathcal{C}(\mathcal{B}_{\mathcal{F}}) + \tfrac{(m-1)n\mu}{\lambda} + \tfrac{(m-2)^2}{\sqrt{\lambda}}.
\end{aligned}
$$

The theorem is proved. $\hfill\square$

## D  INFORMATION-THEORETICALLY SAFETY AGAINST MULTI STEP ATTACK

In this section, we consider muilt-step attacks such as PGD. The original model gradient-based attack with $k$ steps is defined as follows:

**Definition 27.** *Let $\mathcal{F} : \mathbb{I}^n \to \mathbb{R}^m$ be a network and $D_{\mathcal{F}}(x, y) : \mathbb{I}^n \times \mathbb{R} \to \mathbb{R}^n$ be an attack direction depending on sample $(x, y)$, network output $\mathcal{F}(x)$, and gradient $\frac{\nabla \mathcal{F}(x)}{\nabla x}$. $\mathcal{A}_{\mathcal{B}_{\mathcal{F}}, D_{\mathcal{F}}, k} : \mathbb{R}^n \to \mathbb{R}^n$ is called an* original model gradient-based attack *for a bias classifier $\mathcal{B}_{\mathcal{F}}$ based on* attack direction $D_{\mathcal{F}}$ *with $k$ steps if*

$$
\mathcal{A}_{\mathcal{B}_{\mathcal{F}}, D_{\mathcal{F}}, k}(x) = x + \frac{\rho}{k} \sum_{i=1}^{k} D_{\mathcal{F}}(x_i), \tag{22}
$$

*where $x_1 = x$ and $x_j = x_{j-1} + \rho\mathcal{D}_{\mathcal{F}}(x_{j-1})$, $j = 2, \cdots, k$.*

The problem with multi-steps attacks is that the adversarial noise $\delta$ found by the multi-steps attack may not meet the conditions $||\delta||_\infty = \epsilon$, where $\epsilon$ is the budget of attack. So, we define that $\mathcal{C}_1(\mathcal{B}_{\mathcal{F}}) = \frac{1}{2^n} \sum_{V \in \{-1, 1\}^n} \mathbb{E}_{x \sim \mathcal{D}}[\mathbf{I}(\widehat{\mathcal{B}}_{\overline{\mathcal{F}}}(x + \rho_1 V) \neq \widehat{\mathcal{B}}_{\overline{\mathcal{F}}}(x), \forall |\rho_1| \leq \rho)]$, and the $\mathcal{B}_{\overline{\mathcal{F}}}$ which is defined in equation 12 is called **information-theoretically safe** against $\mathcal{A}_{\mathcal{B}_{\overline{\mathcal{F}}}, \mathcal{D}_{\overline{\mathcal{F}}}, k}$, if

$$
\mathcal{C}(\mathcal{B}_{\overline{\mathcal{F}}}, \mathcal{A}_{\mathcal{B}_{\overline{\mathcal{F}}}, \mathcal{D}_{\overline{\mathcal{F}}}, k}, \mathcal{M}) \leq \mathcal{C}_1(\mathcal{B}_{\mathcal{F}}). \tag{23}
$$

We will demonstrate that multi-steps attacks are also information-theoretically safety for an appropriate distribution of $W_R$, to be defined as follows.

**Definition 28.** *Let $\mathcal{V}_{m,n}(\lambda)$ be the distribution in $\mathbb{R}^{m \times n}$ constructed as follows.*

*(1) Randomly select a vector $V$ in $\{-1, 1\}^n$;*

*(2) Elements of the $i$-th row of $\mathcal{V}_{m,n}(\lambda)$ are randomly selected in $\mathcal{I}((2i-1)\lambda, 2i\lambda)^n \circ V$, where $\circ$ is the element-wise product.*

Now we will show that, with this distribution and attack direction equation 14, the bias classifier is information-theoretically safety.

**Theorem 29.** *For $\lambda \in \mathbb{R}_+$, if $||J_{\mathcal{F}}||_\infty < \lambda/2$, $W_R \sim \mathcal{V}_{m,n}(\lambda)$, and $\mathcal{D}_{\mathcal{F}}^1$ is defined in equation 14, then $\mathcal{B}_{\overline{\mathcal{F}}}$ is information-theoretically safe against the attack $\mathcal{A}_{\mathcal{B}_{\overline{\mathcal{F}}}, \mathcal{D}_{\mathcal{F}}^1, k}$.*

*Proof.* Similar to the proof of equation equation 15, we have

$$
\begin{aligned}
\mathcal{A}_{\mathcal{B}_{\overline{\mathcal{F}}}, \mathcal{D}_{\mathcal{F}}^1, k} &= x + \frac{\rho}{k} \sum_{i=1}^{k} \mathrm{sign}(\tfrac{\nabla \overline{\mathcal{F}}_{n_{x_i}}(x_i)}{\nabla x_i} - \tfrac{\nabla \overline{\mathcal{F}}_{y_i}(x_i)}{\nabla x_i}) \\
&= x + \frac{\rho}{k} \sum_{i=1}^{k} \mathrm{sign}(W_{x_i, n_{x_i}} - W_{x_i, y_i} + W_{R, n_{x_i}} - W_{R, y_i}) \\
&= x + \frac{\rho}{k} \sum_{i=1}^{k} \mathrm{sign}(W_{R, n_{x_i}} - W_{R, y_i}).
\end{aligned} \tag{24}
$$

Since $W_R \in \mathcal{V}_{m,n}(\lambda)$, we have $\mathrm{sign}(W_{R, n_{x_i}} - W_{R, y_i}) = V$ or $\mathrm{sign}(W_{R, n_{x_i}} - W_{R, y_i}) = -V$, where $V$ is the random vector in Definition 28. Thus, for some $s \in [0, k]$, we have $\mathrm{sign}(\sum_{i=1}^{k} \mathrm{sign}(W_{R, n_{x_i}} - W_{R, y_i})) = \mathrm{sign}(sV) = sV$.

So we have that:

$$
\begin{aligned}
& \mathcal{C}(\mathcal{B}_{\overline{\mathcal{F}}}, \mathcal{A}_{\mathcal{B}_{\overline{\mathcal{F}}}, \mathcal{D}_{\overline{\mathcal{F}}}, k}, \mathcal{M}) \\
=\ & \mathbb{E}_{W_R \sim \mathcal{M}}[\mathbb{E}_{x \sim \mathcal{D}}[\mathbf{I}(\widehat{\mathcal{B}}_{\overline{\mathcal{F}}}(\mathcal{A}_{\mathcal{B}_{\overline{\mathcal{F}}}, \mathcal{D}_{\overline{\mathcal{F}}}, k}(x)) \neq \widehat{\mathcal{B}}_{\overline{\mathcal{F}}}(x))]] \\
=\ & \mathbb{E}_{V \in \{-1,1\}^n}[\mathbb{E}_{x \sim \mathcal{D}}[\mathbf{I}(\widehat{\mathcal{B}}_{\mathcal{F}}(x + s_x V) \neq \widehat{\mathcal{B}}_{\mathcal{F}}(x))]] \\
\leq\ & \frac{1}{2^n} \sum_{V \in \{-1,1\}^n} \mathbb{E}_{x \sim \mathcal{D}}[\mathbf{I}(\widehat{\mathcal{B}}_{\mathcal{F}}(x + \rho_1 V) \neq \widehat{\mathcal{B}}_{\mathcal{F}}(x), \forall |\rho_1| \leq \rho)]
\end{aligned}
$$

Where $s_x$ is a value in $[-\rho, \rho]$, as shown in the above. This is what we want, so the theorem is proved. $\qquad\square$

If with attack direction equation 16 and $m = 2$, the bias classifier is also information-theoretically safety.

**Theorem 30.** *For $m = 2, \lambda \in \mathbb{R}_+$, if $\|J_{\mathcal{F}}\|_\infty < \lambda/2$, $W_R \sim \mathcal{V}_{m,n}(\lambda)$, and $\mathcal{D}_{\mathcal{F}}^2$ is defined in equation 16, then $\mathcal{B}_{\overline{\mathcal{F}}}$ is information-theoretically safe against attack $\mathcal{A}_{\mathcal{B}_{\overline{\mathcal{F}}}, \mathcal{D}_{\mathcal{F}}^2, k}$.*

*Proof.* The theorem can be proved by combing the proofs of Theorems 14 and 29.

At last, we show that $\mathcal{C}_1(\mathcal{B}_{\mathcal{F}})$ is also very small. Following the section 5 and similar as table 3, the value of $\mathcal{C}_1(\mathcal{B}_{\mathcal{F}})$ is shown in the below:

Table 7: Value of $\mathcal{C}_1(\mathcal{B}_{\mathcal{F}^{(1)}}), \mathcal{C}_1(\mathcal{B}_{\mathcal{F}^{(2)}})$.

|  | MNIST | CIFAR-10 |
|---|---|---|
| $\mathcal{C}(\mathcal{F}^{(1)})$ | 1.02% | 1.30% |
| $\mathcal{C}(\mathcal{F}^{(2)})$ | 1.04% | 1.42% |

Compare with table 3, the value of $\mathcal{C}_1$ is about $0.2\%$ larger than $\mathcal{C}$.

# E  MORE EXPERIMENTS AND DETAIL

Following the section 5.1, some results under FGSM attack and AutoAttack are given.

Table 8: Adversarial accuracy with the original model gradient-based attack(FGSM).

| Network | MNIST | CIFAR-10 | Network | MNIST | CIFAR-10 |
|---|---|---|---|---|---|
| $\mathcal{B}_{\mathcal{F}^{(5,1)}}$ | 98.70% | 80.16% | $\mathcal{B}_{\mathcal{F}^{(5,2)}}$ | 98.60% | 80.05% |
| $\mathcal{B}_{\mathcal{F}^{(6,1)}}$ | 98.73% | 80.85% | $\mathcal{B}_{\mathcal{F}^{(6,2)}}$ | 98.77% | 80.63% |
| $\mathcal{B}_{\mathcal{F}^{(7,1)}}$ | 98.30% | 79.42% | $\mathcal{B}_{\mathcal{F}^{(7,2)}}$ | 98.45% | 79.46% |

Table 9: Adversarial accuracy with the original model gradient-based attack(Auto Attack but remove the black box attack from it).

| Network | MNIST | CIFAR-10 | Network | MNIST | CIFAR-10 |
|---|---|---|---|---|---|
| $\mathcal{B}_{\mathcal{F}^{(5,1)}}$ | 98.64% | 80.13% | $\mathcal{B}_{\mathcal{F}^{(5,2)}}$ | 98.47% | 80.03% |
| $\mathcal{B}_{\mathcal{F}^{(6,1)}}$ | 98.67% | 80.87% | $\mathcal{B}_{\mathcal{F}^{(6,2)}}$ | 98.72% | 80.60% |
| $\mathcal{B}_{\mathcal{F}^{(7,1)}}$ | 98.20% | 79.39% | $\mathcal{B}_{\mathcal{F}^{(7,2)}}$ | 98.41% | 79.32% |

It is easy to see that bias classifier has almost achieve Information-theoretically safety under FGSM and AutoAttack.

$\qquad\square$

