# OpenReview forum: "Information-theoretically Safe Bias Classifier Against Adversarial Attacks"
_ICLR.cc/2025/Conference — ICLR 2025 Conference Withdrawn Submission_

### Official Review · Reviewer_brpS · 2024-10-30

**Soundness:** 2
**Presentation:** 1
**Contribution:** 2
**Rating:** 3
**Confidence:** 4

**Summary:**

The paper proposes a bias classifier and proofs that it serves as a universal classifier. Additionally, it introduces a new concept called "information-theoretically safe." Finally, based on experimental results, the authors claim that the adversarially trained bias classifier is information-theoretically safe against various white-box and black-box attack methods.

**Strengths:**

The paper proposes and introduces a new adversarial-related definition called "information-theoretically safe" along with a new model termed "bias classifier," which may be valuable to the literature.

**Weaknesses:**

I strongly recommend that the author include a notation section in the revised version.

Here, I have listed several questions that arose while reviewing the paper.

Line 060-061, What is meant by a "random direction attack"? Does this refer to adding random noise to the DNN input? The motivation behind this new definition is unclear. The upper bound of adversarial accuracy for a classifier can be defined as its classification accuracy on clean examples. What advantage does the paper's definition offer over this approach? Additionally, if a DNN is highly sensitive to noise, achieving zero accuracy on any noisy input, it would trivially meet the "information-theoretically safe" criterion, despite having no robustness at all.


Line 063, "By the bias classifier, we mean to use $\mathcal{F}$’s bias part ${\mathcal{B_F}}(x)$  = $\mathcal{B}x$ : $\mathbb{I}^n \rightarrow R$ m as a classifier." A classifier is a function that maps the input $x$ to a output prediction. Could the authors clarify how the bias term of a classifier can itself function as a classifier? $\mathbb{I} = [0, 1]$. What is $\mathbb{I} $? Does [0, 1] means $\mathbb{I}$ is bounded between 0 and 1?

Line 099-100, [1] is a review paper that does not focus on DNN structure, so why is it cited here?

The related work section is somewhat disorganized. The authors just list different approches without detailed explaination. Further, Madry's adversarial training is well-known in the literature, it is not considered one of the most effective practical defense methods from nearly all perspectives.

Line 119, Where is the new loss function?

Line 175-177, "The following existence theorem shows that the bias classifier has the power to interpolate any classification functions with arbvitray high probability". Minor: arbitrary is misspelled.

Section 3.3 What's the motivation of using adversarial training to train the bias classifier?

Line 229-230.
Gradient-based adversarial attacks are well-established in the literature and have been studied for years; thus, the concept is not original to the author.

Line 249.
What does the author mean by stating that a neural network involves a random variable? It is more common to say that a random variable follows a specific distribution.


Line 259, Could the authors further clarification on the formula of $\mathcal{C}(\mathcal{F}_R)$?

Equation (14) seems wrong.

Line 379-380, What is the "original based AutoAttack"?

Line 480-481, "bias classifier is essentially different from gradient obfuscated defences." I don't believe it is appropriate to make such a strong statement based on experiments involving only one method published in 2018. Could the author provide more context or additional evidence?

The experimental results are insufficient. The authors did not benchmark and compare against the state-of-the-art methods and only test their method on small datasets like MNIST and CIFAR-10.

[1] Han Xu, Yao Ma, Hao-Chen Liu, Debayan Deb, Hui Liu, Ji-Liang Tang, and Anil K Jain. Adversarial attacks and defenses in images, graphs and text: A review. International journal of automation and computing, 17:151–178, 2020.

**Questions:**

Please refer to the weakness.

---

### Official Review · Reviewer_jVkx · 2024-11-03

**Soundness:** 2
**Presentation:** 2
**Contribution:** 2
**Rating:** 3
**Confidence:** 4

**Summary:**

This paper proposes to adversarially train the model and the bias and then take the bias as the classifier (dubbed as bias classifier), which improves the adversarial robustness. The authors theoretically show that the bias classifier can be information-theoretically safe against certain kinds of adversarial attacks, i.e., FGSM and CW attacks. Experimental results using the ResNet and VGG validate the effectiveness of the bias classifier.

**Strengths:**

1.	It is interesting to study the potential of utilizing the bias term of a neural network towards improved adversarial robustness. It seems that the authors theoretically show the bias classifier can defend against certain kinds of adversarial attacks.
2.	The empirical results seem to support the claim.

**Weaknesses:**

1.	The scope of the theory is limited. The authors only show the potential theoretically-information safety of the bias classifier against two certain adversarial attacks, i.e., FGSM and CW attacks, instead of the generalized adversarial attacks.
2.	The experiments are conducted on simple networks and small datasets, which cannot well support the effectiveness of the bias classifier.
3.	The presentation is somewhat unclear. For example, what does ‘the original model gradient-base attacks’ refer to in Table 5? Besides, in Table 5, which results refers to the baseline?

**Questions:**

Please refer to Weakness.

---

### Official Review · Reviewer_jt8H · 2024-11-03

**Soundness:** 1
**Presentation:** 2
**Contribution:** 2
**Rating:** 1
**Confidence:** 4

**Summary:**

The paper proposes Bias Classifiers which results from subtracting the linear approximation of a ReLU activated neural network on some test sample from the network. The paper is mostly focused on proving that if we assume that the norm of the Jacobian is less than some upper bound, then we can infer that the bias classifier is robust.

**Strengths:**

- Originality:

The paper is somewhat original in considering test time training of a classifier.

- Quality:

The paper is well-written for the most part.

- Clarity:

The paper uses easy to follow instructions.

- Significance:

The paper considers a significant problem.

**Weaknesses:**

The paper is not ready for publication for various reasons. Here are some key observations:

1) The paper does not make use of standard notation, e.g. $\frac{\nabla F(x)}{\nabla x}$ is not meaningful as it seems to be a division of two vectors. Maybe the authors meant to write $\frac{\nabla F(x)}{|\nabla F(x)|}$?

2) The paper is testing a self-fulfilling prediction where an accurate classifier over benign test samples is forced to be constant on a neighborhood of the benign test sample and then finding out that it is both robust and accurate. A real test of robustness of this method should involve linearization on an adversarial test sample for which the reference network is *not* accurate.

3) The idea of information theoretic safety is very similar the idea of probabilistic robustness which is already discussed in the literature. Moreover, the description is vague, e.g. "Let $\mathcal{F}_R$ be a neural network that involves a random variable $R$ satisfying the distribution $\mathcal{R}$". What does it mean for a neural network to involve a random variable?

**Questions:**

See weaknesses.

---

### Official Review · Reviewer_QJsc · 2024-11-10

**Soundness:** 1
**Presentation:** 1
**Contribution:** 1
**Rating:** 1
**Confidence:** 5

**Summary:**

This paper designs a "provable" adversarial robustness enhancing method named "bias classifier" that is supposed to help ML models achieve high adversarial accuracy. A new information-theoretically inspired notation is proposed to measure the provable robustness of the bias classifier method. Experiments are conducted on two simple datasets, i.e., MNIST and CIFAR-10, and the results show that the adversarial robustness of models obtained from the bias classifier method is extremely weak (compared with existing adversarial robustness approaches).

**Strengths:**

None.

**Weaknesses:**

This is a wrong paper and should be rejected immediately. Detailed comments are as follows:

1. The authors claim that their proposed bias classifier method has provable high adversarial robustness. However, according to the experiment results in Section 5, the robustness of the proposed method is actually extremely WEAK compared with existing adversarial robustness approaches. Specifically:

    - In Table 2, the proposed method can only achieve an adversarial accuracy (adversarial radius $\rho = 8/255$) of $31.25%%$ on the CIFAR-10 dataset under the AutoAttack [r1]. However, according to the original AutoAttack paper [r1], under the same setting, the best adversarial accuracy is $59.53%%$ in 2020 (see Table 2 in [r1]), which is almost two times larger than the performance of the proposed bias classifier method.

    - In Table 5, although the adversarial accuracies seems to be high, they are actually calculated based on adversarial examples crafted with **random noise** (see Line#394-400). Such an attack is even much weaker than the simplest FGSM attack. Thus, Table 5 is meaningless and does not show any advantages of the proposed bias classifier method.

2. The authors claim that existing defenses cannot give provable adversarial robustness (see Line#050). Unfortunately, this is a **wrong claim** since the well-known "certified robustness" method [r2] can indeed give the model provable and high adversarial robustness. For example, [r3] shows that with the help of diffusion models, when the adversarial perturbation is not stronger than $127/255$ under $l\_2$-norm, one can provablely ensure an adversarial accuracy of $65.5\\%$ on the CIFAR-10 dataset. [r4] further improves such an provable adversarial accuracy to $70.7\\%$. Given the profound success of existing certified robustness methods, I am wondering what is the advantage of the proposed bias classifier method compared with existing approaches.

3. The overall goal of the bias classifier method is to ensure ML models satisfy Eq.(11). Unfortunately, satisfying Eq.(11) does not mean that the model indeed enjoys a **provablely high** robustness. This is because Eq.(11) only ensures that small perturbations would not change model predictions, but the model clean accuracy could be very low in order to fulfill the requirement of Eq.(11). Actually, experiments in Section 5 indeed show that models obtained from bias classifier method and satisfying Eq.(11) indeed enjoys a very weak clean accuracy (since low robust accuracy implicitly imply low clean accuracy according to this paper). I suggest the authors read more papers about certified robustness, which can achieve "**provablely high** adversarial robustness".

**References:**

[r1] Croce F. and Hein M. "Reliable Evaluation of Adversarial Robustness with an Ensemble of Diverse Parameter-free Attacks." ICML 2020.

[r2] Cohen J. et al. "Certified Adversarial Robustness via Randomized Smoothing." ICML 2019.

[r3] Carlini N. et al. "(Certified!!) Adversarial Robustness for Free!" arXiv 2206.10550 / ICLR 2023.

[r4] Chen H. et al. "Diffusion Models are Certifiably Robust Classifiers." arXiv 2402.02316 / NeurIPS 2024.

**Questions:**

None.

---

### Note · Authors · 2024-11-21

I have read and agree with the venue's withdrawal policy on behalf of myself and my co-authors.